# Task Generalization With AutoRegressive Compositional Structure: Can Learning From $D$ Tasks Generalize to $D^T$ Tasks?

Amirhesam Abedsoltan [* 1]  Huaqing Zhang [* 2]  Kaiyue Wen [3]  Hongzhou Lin [4]  Jingzhao Zhang [2]
Mikhail Belkin [1 5]

## Abstract

Large language models (LLMs) exhibit remarkable task generalization, solving tasks they were never explicitly trained on with only a few demonstrations. This raises a fundamental question: When can learning from a small set of tasks generalize to a large task family? In this paper, we investigate task generalization through the lens of autoregressive compositional structure, where each task is a composition of $T$ operations, and each operation is among a finite family of $D$ subtasks. This yields a total class of size $D^T$. We first show that generalization to all $D^T$ tasks is theoretically achievable by training on only $\tilde{O}(D)$ tasks. Empirically, we demonstrate that Transformers achieve such exponential task generalization on sparse parity functions via In-context Learning (ICL) and chain-of-thought (CoT) reasoning. We further show generalization in arithmetic and translation, beyond parity functions.

## 1. Introduction

Large language models (LLMs) demonstrate a remarkable ability to solve tasks they were never explicitly trained on. Unlike classical supervised learning, which typically assumes that the test data distribution follows the training data distribution, LLMs can generalize to new task distributions with just a few demonstrations—a phenomenon known as in-context learning (ICL) (Brown et al., 2020; Wei et al., 2022; Garg et al., 2022; He et al., 2024; Lee et al., 2023). Recent studies suggest that trained Transformers implement algorithmic learners capable of solving various statistical

---

[*]Equal contribution  [1]Department of Computer Science and Engineering, UC San Diego. [2]Institute for Interdisciplinary Information Sciences, Tsinghua University. [3]Stanford University. [4]Amazon. This work is independent of and outside of the work at Amazon. [5]Halicioglu Data Science Institute, UC San Diego. Correspondence to: Amirhesam Abedsoltan <aabedsoltan@ucsd.edu>.

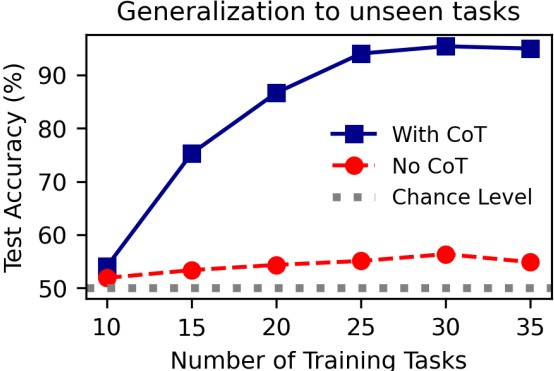

*Figure 1.* We train a Transformer to learn parity functions through In-Context Learning (ICL), where all demonstrations and the query are concatenated into a single input prompt. Each prompt corresponds to one specific target function $f_i$ and contains binary input–output pairs $(\mathbf{x}_1, f_i(\mathbf{x}_1)), \ldots, (\mathbf{x}_n, f_i(\mathbf{x}_n))$ followed by a query $\mathbf{x}_{\text{query}}$, with $\mathbf{x}_j \in \{0,1\}^d$. The functions $f_i$ are drawn from the $\text{Parity}(10, 2)$ family ($d = 10$, $k = 2$), yielding 45 possible tasks. For testing, 10 such functions were withheld entirely, and different subsets of the remaining ones were used for training. Consistent with prior work (Bhattamishra et al., 2024), we observe that standard ICL fails to generalize across tasks. In contrast, incorporating Chain-of-Thought (CoT) reasoning significantly improves performance on unseen tasks.

tasks—such as linear regression—at inference time in context (Li et al., 2023; Bai et al., 2023).

Despite their success in tasks such as learning conjunctions or linear regression, Transformers relying solely on ICL struggle with other problems, particularly those requiring hierarchical reasoning. A notable case where Transformers fail is out-of-domain learning of sparse parity functions, as examined in (Bhattamishra et al., 2024). During training, a Transformer is exposed to data $(\mathbf{x}_1, f_i(\mathbf{x}_1)), \ldots, (\mathbf{x}_n, f_i(\mathbf{x}_n))$ generated from multiple target functions $f_i, i \in \mathcal{I}$, each defining a distinct sparse parity task. In testing, data from a different task $f_j, j \notin \mathcal{I}$ is given in-context and evaluated on a new data point. The results in (Bhattamishra et al., 2024) show that Transformers fail to generalize to these unseen functions, even when each tasks is a simple XOR operation on two bits of the binary input

string.

In contrast, we found that incorporating Chain-of-Thought (CoT) reasoning — adding intermediate reasoning steps to the prompt — allows Transformers to easily generalize to unseen tasks. This is illustrated in Figure 1. Consistent with (Bhattamishra et al., 2024), we observe that Transformers without CoT perform only slightly better than chance level, no matter how many training tasks are presented to the model. Conversely, as the number of training tasks increases, Transformers with CoT achieve near-perfect generalization on the held-out set of unseen tasks. We see that the extra information provided by CoT enables the model to exploit the compositional structure of the parity problem.

Motivated by this example, we aim to systematically analyze how models can leverage autoregressive compositional structures to extend their capabilities beyond the training tasks. Conventionally, learning involves approximating a target function $f^*$ drawn from a function class $\mathcal{F}$ using examples from a training distribution over the input space $\mathcal{X}$; generalization is then measured by testing $f^*$ on new examples. In contrast, our focus is on **task generalization**, where training is restricted to a subset of functions or "tasks" $\mathcal{F}_{\text{train}} \subset \mathcal{F}$, leaving the remaining functions, unseen during training. Our goal is to investigate whether a model trained on tasks from $\mathcal{F}_{\text{train}}$ (with inputs from $\mathcal{X}$) can generalize to *all tasks*, including *unseen* tasks.

This notion of task generalization goes beyond the standard out-of-distribution (OOD) settings (see, e.g., (Zhou et al., 2022) for review) by shifting the focus from adapting to new input distributions to learning entirely new tasks. Specifically, we ask:

*How can we quantify the number of tasks a model must be trained on to generalize to the entire class $\mathcal{F}$?*

To analyze task generalization, we consider a finite set of functions $\mathcal{F}$, where each function maps an input $\boldsymbol{x} \in \mathcal{X}$ to a tuple of random variables $\boldsymbol{y} = (y_1, \dots, y_T)$. We assume each function can be characterized by a parameter tuple $\theta = (\theta_1, \theta_2, \dots, \theta_T)$. The outputs are generated autoregressively: first, $y_1$ is produced from $\boldsymbol{x}$; then $y_2$ is generated from $\boldsymbol{x}$ and $y_1$; and then $y_3$ is generated from $\boldsymbol{x}$, $y_1$ and $y_2$; and this process continues until $y_T$ is produced. Specifically, the sequence is generated sequentially as:

$$y_t \sim P_{\theta_t}(y_t \mid \boldsymbol{x}, \boldsymbol{y}_{<t}), \quad \text{for } t = 1, \dots, T,$$

where $\boldsymbol{y}_{<t} = (y_1, \dots, y_{t-1})$ denotes the previously generated outputs, and $P_{\theta_t}$ is some conditional probability distribution that is parametrized by $\theta_t$ and is conditioned on $\boldsymbol{y}_{<t}$ and $\boldsymbol{x}$. This structure can also be interpreted as a sequence of compositions,

$$\boldsymbol{x} \xrightarrow{P_{\theta_1}} y_1$$
$$\boldsymbol{x}, y_1 \xrightarrow{P_{\theta_2}} y_2$$
$$\dots$$
$$\boldsymbol{x}, y_1, \dots, y_{T-1} \xrightarrow{P_{\theta_{T-1}}} y_T \ .$$

We will call this function class *AutoRegressive Compositional structure*. Assuming that the cardinality of the set of possible values for each parameter $\theta_t$ is finite and is equal to $D$, we will use the notation $\mathcal{F} = ARC(T, D)$. The cardinality of this class is $D^T$.

For the sparse parity problem with $k$ secret keys in this framework, the output sequence has length $T = k$. As an example, consider $n = 5$ and $k = 3$ with the secret key set $S = \{1, 2, 4\}$. Here, the inputs $\boldsymbol{x} = (x_1, x_2, \dots, x_n)$ are binary sequences of length $n$, and the target function is $f_S(x_1, x_2, x_3, x_4, x_5) = x_1 \oplus x_2 \oplus x_4$, where $\oplus$ denotes bitwise XOR. The computation can be expressed step-by-step as:

$$\underbrace{y_1 = x_1}_{\text{Step 1}} \quad \underbrace{y_2 = x_1 \oplus x_2}_{\text{Step 2}} \quad \underbrace{y_3 = x_1 \oplus x_2 \oplus x_4}_{\text{Step 3}}.$$

That is, each $y_t$ recovers the XOR of the first $t$ secret coordinates. For example, if the input sequence is 10101, the corresponding chain-of-thought will be 111. In this case, the output distribution at each step is deterministic, assigning probability 1 to the correct XOR value and 0 to other values.

We can now address the following fundamental question:

*How many tasks in $\mathcal{F}_{\text{train}}$ must a model be trained on to generalize to all tasks in $\mathcal{F}$, including those it has not seen? In particular, can a model trained on $\tilde{O}(D)$ tasks generalize across the entire set of $D^T$ tasks?*

Our main contributions are as follows:

- We define AutoRegressive Compositional structure and introduce a framework to quantitatively analyze task generalization when the function class follows an AutoRegressive Compositional structure. (Sections 3.2 and 3.3)

- We establish that under this structure, task generalization to all $D^T$ tasks is theoretically achievable by training on $\tilde{O}(D)$ tasks up to logarithmic terms (Section 3.4).

- We demonstrate how the parity problem aligns with our framework and empirically show that Transformers trained on i.i.d. sampled tasks exhibit exponential task generalization via chain-of-thought (CoT) reasoning, consistent with theoretical scaling (Section 4).

- Finally, we show that the selection of training tasks significantly impacts generalization to unseen tasks. If tasks are chosen adversarially, training on even nearly all $D^T$ of the tasks with CoT may fail to generalize to the remaining tasks (Section 5.1).

## 2. Related Works

### 2.1. Composition and Generalization

The role of composition in reasoning for language models has been widely studied. (Saparov et al., 2023) explores various out-of-distribution (OOD) generalization formats, including compositional generalization, showing that a neural network's ability to generalize compositionally is highly dependent on both architecture and task properties. Similar conclusions have been drawn in prior works (Lake & Baroni, 2018; Keysers et al., 2020). Further, (Bhattamishra et al., 2024; Dziri et al., 2023; An et al., 2023; Xu et al., 2024) examine compositional generalization in in-context learning (ICL) and find that generalization to composing multiple steps is in general hard for LLMs. One notable observation is that LLMs succeed in compositional generalization for clause satisfaction problems but not for parity problems.

Another line of research investigates composition as a mechanism underlying emergent abilities in language models. (Arora & Goyal, 2023) demonstrates that language modeling can lead to learning tuples of skills, which are small compositions of fundamental capabilities. Building on this idea, (Kaur et al., 2025; Zhao et al., 2024) leverage compositional structures to generate supervised fine-tuning (SFT) data, leading to improved language model performance.

Beyond sequential composition, other forms of compositionality in neural networks have been explored. (Song et al., 2025) investigates layer-wise composition in transformers, while (Schug et al., 2024) proposes a modular neural architecture for learning hidden compositional representations. Additionally, (Wiedemer et al., 2024) examines compositional structures in image reconstruction, and (Lippl & Stachenfeld, 2025) provides a theoretical analysis of composition in kernel and linear models.

While prior work has largely focused on qualitative insights into compositional generalization, our work takes a quantitative approach: studying how many training tasks are needed to achieve task generalization over an entire function class.

### 2.2. Learning and Testing with Multiple Distributions

Our work aims to analyze generalization when the training and testing distributions differ. This problem has been studied from various perspectives in the statistical learning community. One approach is to frame it as learning a shared representation across multiple tasks. (Ye et al., 2021) defines variation and informativeness between different environments based on a common representation, while (Arjovsky et al., 2019) addresses the problem by designing specific training objectives. Earlier studies on linear and kernel models also explore this direction (Du et al., 2017; Lei et al., 2021).

Another perspective considers the testing environment as a distribution shift, where the model may sample during inference to achieve domain adaptation. (Mansour et al., 2009) analyzes generalization error when a model is trained on distribution $P$ but tested on a different distribution $Q$, introducing an error bias dependent on the distance $d(P, Q)$. To mitigate this bias, (Cortes et al., 2010) proposes reweighting training samples when test samples from $Q$ are available.

A related line of research investigates scenarios where both training and test samples are accessible. Notable setups include covariate shift (Kpotufe & Martinet, 2021; Ma et al., 2023) and domain adaptation (Sugiyama et al., 2007; Ben-David & Urner, 2014). When direct sampling from the test distribution is not feasible, alternative strategies focus on training robustly against worst-case shifts. This can be achieved through adversarial perturbations or min-max optimization formulations (Madry et al., 2018; Raghunathan et al., 2020; Duchi et al., 2023).

In this work we impose an AutoRegressive Compositional(ARC) structure on the function class and propose a new framework to study task generalization. This compositional structure decomposes the function class into atomic subtasks, enabling a modular approach to learning. By leveraging this structure, we establish a quantitative understanding of how many training tasks are required for generalization. Our results provide a theoretical foundation for structured learning and demonstrate how models can efficiently generalize beyond the training distribution.

## 3. Theoretical Framework for Task Generalization

In this section, we present a theoretical framework to study task generalization with autoregressive compositional structure. When the compositional structure holds, we show that there exists a learning algorithm that is only trained on $\tilde{O}(D)$ different tasks, but can generalize to exponentially many unseen tasks.

### 3.1. Preliminaries and Notations

For a positive integer $n$, denote $[n] = \{1, 2, \cdots, n\}$. For a finte set $\mathcal{S}$, we denote by $\Delta(\mathcal{S})$ the probability simplex over with support $\mathcal{S}$. Given $t$ sets $\mathcal{S}_1, \mathcal{S}_2, \ldots, \mathcal{S}_t$, their **Cartesian product** is defined as

$$\underset{i=1}{\overset{t}{\bigtimes}} \mathcal{S}_i := \{(s_1, s_2, \ldots, s_t) \mid s_i \in \mathcal{S}_i \text{ for all } i \in [t]\}.$$

We further denote $\mathcal{S}^t := \bigtimes_{i=1}^t \mathcal{S}$. For two probability distributions $P$ and $Q$ over a discrete space $\mathcal{S}$, the **total variation distance** is defined as

$$\text{TV}(P, Q) := \frac{1}{2} \sum_{s \in \mathcal{S}} |P(s) - Q(s)|.$$

We let the bold letter $\boldsymbol{y}$ denote a sequence, and the subscripted $\boldsymbol{y}^j$ denote the $j_{th}$ sequence / example. Within each sequence $\boldsymbol{y}^j = (y_1, ..., y_T)$, the regular letter $y_t$ denote the $t_{th}$ token in the sequence.

### 3.2. AutoRegressive Compositional Structure

In the following definition, we formally introduce the **AutoRegressive Compositional (ARC)** task class, which models structured sequence generation through a composition of conditional distributions.

**Definition 3.1.** *(AutoRegressive Compositional task class).* Let $\mathcal{X}$ and $\mathcal{Y}$ denote the finite input and output spaces, respectively. The AutoRegressive Compositional (ARC) task class consists of sequential generation processes:

$$\mathcal{F} := \{f_\theta = (P_{\theta_1}, \ldots, P_{\theta_T}) \mid P_{\theta_t} \in \mathcal{P}_{\Theta_t} \text{ for all } t \in [T]\},$$

where each task $f_\theta \in \mathcal{F}$ for any input $\boldsymbol{x} \in \mathcal{X}$ generates an output sequence $\boldsymbol{y} = (y_1, \ldots, y_T) \in \mathcal{Y}$ through an autoregressive sampling process:

$$y_t \sim P_{\theta_t}(\cdot \mid \boldsymbol{x}, \boldsymbol{y}_{<t}), \quad \text{for all } t \in [T].$$

At each step $t$, the conditional probability distribution $P_{\theta_t}$ is drawn from a subtask family $\mathcal{P}_{\Theta_t}$, parametrized by $\theta_t$:

$$\mathcal{P}_{\Theta_t} := \{P_{\theta_t}(\cdot \mid \boldsymbol{x}, \boldsymbol{y}_{<t}) : \mathcal{X} \times \mathcal{Y}^{t-1} \to \Delta(\mathcal{Y}) \mid \theta_t \in \Theta_t\}.$$

Here, $\Theta_t$ represents the parameter space at step $t$, and the overall task parameter space is $\Theta := \bigtimes_{t=1}^T \Theta_t$. Assuming each step has a finite number of possible subtasks, i.e., $|\Theta_t| = d$ for all $t \in [T]$, the AutoRegressive Compositional task class $ARC(d, T)$ consists of $|\mathcal{F}| = |\Theta| = d^T$ tasks.

Given any input $\boldsymbol{x} \in \mathcal{X}$ and a sequence $\boldsymbol{y} \in \mathcal{Y}^T$, the joint distribution for a task $f_\theta = (P_{\theta_1}, \cdots, P_{\theta_T}) \in \mathcal{F}$ is:

$$P_\theta(\boldsymbol{x}, \boldsymbol{y}) = P(\boldsymbol{x}) \prod_{s=1}^T P_{\theta_s}(y_s \mid \boldsymbol{x}, \boldsymbol{y}_{<s}), \quad (1)$$

and for partial sequences up to any $t \in [T]$:

$$P_{\theta_{1:t}}(\boldsymbol{x}, \boldsymbol{y}_{1:t}) = P_x(\boldsymbol{x}) \prod_{s=1}^t P_{\theta_s}(y_s \mid \boldsymbol{x}, \boldsymbol{y}_{<s}). \quad (2)$$

At a high level, an AutoRegressive Compositional task class $ARC(D, T)$ is characterized by two key properties:

- **Modularity.** The generation process is decomposed into $T$ sequential steps, each governed by an independent conditional distribution $P_{\theta_t} \in \mathcal{P}_{\Theta_t}$. This modular structure allows tasks to be constructed by combining different components at each step.

- **Exponential Growth.** The task class size grows exponentially in $T$ as $|\mathcal{F}| = D^T$, despite each step having only $D$ choices. This reflects the combinatorial nature of task construction, where variations at each step lead to an exponentially large set of possible tasks.

### 3.3. Task Generalization

Under the autoregressive task learning setup, there are two levels of generalization:

1. Generalizing to unseen inputs within a task.

2. Generalizing to unseen tasks in the class $\mathcal{F}$.

We focus on the latter one, referred as **task generalization**.

**Training Phase** During training, the model can only access to a small subset of tasks $\mathcal{F}_{\text{train}} = \{f_{\theta^1}, \ldots, f_{\theta^{n_\theta}}\} \subseteq \mathcal{F}$ with $n_\theta = |\mathcal{F}_{\text{train}}|$. For each task $f_{\theta^i} \in \mathcal{F}_{\text{train}}$, we observe $n_x$ i.i.d. demonstration samples:

$$\mathcal{D}_i = \left\{(\boldsymbol{x}^{i,j}, \boldsymbol{y}^{i,j})\right\}_{j=1}^{n_x} \overset{\text{i.i.d.}}{\sim} P_{\theta^i}(\boldsymbol{x}, \boldsymbol{y}),$$

where $P_{\theta^i}$ is defined as in Equation (1). The full training dataset is the union of $\mathcal{D}_i$ denoted by $\mathcal{D}_{\text{train}} = \{\mathcal{D}_i\}_{i=1}^{n_\theta}$.

We assume the learner does not know the true subtask conditional distribution families $\{\mathcal{P}_{\Theta_t}\}_{t=1}^T$ a priori. Instead, it accesses to a **larger hypothesis class**:

$$\mathcal{P}_{\Xi_t} := \left\{P_{\zeta_t}(\cdot \mid \boldsymbol{x}, \boldsymbol{y}_{<t}) \mid \zeta_t \in \Xi_t\right\} \supseteq \mathcal{P}_{\Theta_t},$$

where $\Xi_t$ parameterizes the learner's model class at step $t$. The goal of training is to *identify the true subtask families* $\mathcal{P}_{\Theta_t}$ from $\mathcal{P}_{\Xi_t}$ through $\mathcal{D}_{\text{train}}$.

**Inference Phase** At test time, the learner is given $\ell$ inference-time demonstration samples:

$$\mathcal{D}_{\text{infer}} = \{(\tilde{\boldsymbol{x}}^i, \tilde{\boldsymbol{y}}^i)\}_{i=1}^\ell \overset{\text{i.i.d.}}{\sim} P_{\tilde{\theta}}(\boldsymbol{x}, \boldsymbol{y}),$$

where $f_{\tilde{\theta}} \in \mathcal{F}$ is an unseen task. The learner must identify the true conditionals $\{P_{\tilde{\theta}_t}\}_{t=1}^T$ from $\mathcal{P}_{\Theta_t}$ for each step $t$. Formally, the learner $\mathcal{A}$ that is trained on $\mathcal{D}_{\text{train}}$ and given $\mathcal{D}_{\text{infer}}$ as input, produces an output sequence of conditional distributions:

$$\mathcal{A}(\mathcal{D}_{\text{infer}}; \mathcal{D}_{\text{train}}) \in \{(P_{\xi_1}, \cdots, P_{\xi_T}) \mid P_{\xi_t} \in \mathcal{P}_{\Xi_t}, t \in [T]\}.$$

## 3.4. Main Result: Exponential Task Generalization

We now establish our main theoretical result: with the compositional structure in Definition 3.1, a learner can achieve exponential task generalization with only $\tilde{O}(D)$ training tasks. This demonstrates how compositional structure fundamentally reduces the sample complexity of task learning from exponential to polynomial in $D$. Our results hold under the following mild assumptions:

**Assumption 3.2** (Compositional Identifiability). The autoregressive task class $\mathcal{F}$ satisfies:

1. **Finite Subtask Families.** For each $t \in [T]$, the hypothesis class $\mathcal{P}_{\Xi_t}$ is finite and the subtask conditional distribution family $\mathcal{P}_{\Theta_t} \subseteq \mathcal{P}_{\Xi_t}$ has size $|\mathcal{P}_{\Theta_t}| = D$.

2. **Task Identifiability.** For any $t \in [T]$, $\theta_{1:t-1} \in \bigtimes_{s=1}^{t-1} \Theta_s$, and $\theta_t \in \Theta_t$, $\zeta_t \in \Xi_t$, $P_{\zeta_t} \neq P_{\theta_t}$, the induced distributions stasify:

$$\mathrm{TV}\left(P_{\theta_{1:t-1}, \theta_t}, P_{\theta_{1:t-1}, \zeta_t}\right) > 0.$$

Furthermore, for any $t \in [T]$, $\theta_{1:t-1} \in \bigtimes_{s=1}^{t-1} \Theta_s$, and $\theta_t \neq \theta_t' \in \Theta_t$, the induced distributions satisfy:

$$\mathrm{TV}\left(P_{\theta_{1:t-1}, \theta_t}, P_{\theta_{1:t-1}, \theta_t'}\right) \geq c > 0.$$

Under these conditions, we establish our main theorem:

**Theorem 3.3** (**Exponential Task Generalization**). *Let $\mathcal{F}$ be an AutoRegressive Compositional(ARC) task class satisfying Assumption 3.2. Then there exists a learner $\mathcal{A}$ with the following property: if during training, one samples $n_\theta \geq D \ln(100\, D\, T)$ tasks uniformly and independently from $\mathcal{F}$, each provided with $n_x$ i.i.d. demonstration samples as the training dataset, and if at inference one observes $\ell \geq \frac{2 \ln(100\, T\, n_\theta)}{c^2}$ i.i.d. demonstration samples from a previously unseen task $P_{\tilde{\theta}} \in \mathcal{F}$, then*

$$\lim_{n_x \to \infty} \Pr\left[\mathcal{A}(\mathcal{D}_{\mathrm{infer}}; \mathcal{D}_{\mathrm{train}}) \neq (P_{\tilde{\theta}_1}, \ldots, P_{\tilde{\theta}_T})\right] \leq 0.02,$$

*where $\mathcal{D}_{\mathrm{train}}$ and $\mathcal{D}_{\mathrm{infer}}$ denote the training dataset and inference-time demonstration samples respectively, and the probability is taken over the random selection of training tasks $\mathcal{F}_{\mathrm{train}} \subseteq \mathcal{F}$, the training data $\mathcal{D}_{\mathrm{train}}$, and the inference-time demonstration samples $\mathcal{D}_{\mathrm{infer}}$.*

In other words, Theorem 3.3 shows that the learner can generalize from only $\tilde{O}(D)$ tasks to an exponentially large number of unseen tasks, on the order of $D^T$. The learning algorithm $\mathcal{A}$ operates in two stage. In the training stage, it applies a maximum-likelihood estimation (MLE) procedure to $\mathcal{D}_i$ in order to identify the subtasks of the $i$-th training task $(P_{\theta_1^i}, \cdots, P_{\theta_T^i})$. In the inference stage, it then uses a total-variation-based distribution discrimination test (Lemma A.1) on inference-time demonstration samples $\mathcal{D}_{\mathrm{infer}}$ to recover $f_{\tilde{\theta}} = (P_{\tilde{\theta}_1}, \cdots, P_{\tilde{\theta}_T})$. The proof is deferred to Appendix A.1.

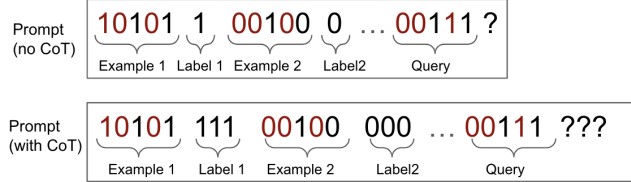

*Figure 2.* Prompt Format Comparison: With vs. Without CoT.

*Remark* 3.4. If we additionally assume that every subtask distribution is separated from any incorrect hypothesis by a fixed total-variation margin, i.e. for all $t \in [T]$, $\theta_{1:t-1} \in \bigtimes_{s=1}^{t-1} \Theta_s$, and $\theta_t \in \Theta_t$, $\zeta_t \in \Xi_t$ with $P_{\zeta_t} \neq P_{\theta_t}$,

$$\mathrm{TV}\left(P_{\theta_{1:t-1}, \theta_t}, P_{\theta_{1:t-1}, \zeta_t}\right) \geq r > 0,$$

then one can replace the MLE procedure used in the training stage with the same distribution-discrimination approach from the inference stage (Lemma A.1). Under this condition, we can derive a *non-asymptotic* bound on the $n_x$ needed per task for accurate identification. See Appendix B for details.

## 3.5. Example: Sparse Parity Problem

To illustrate the role of the AutoRegressive Compositional structure, we use sparse parity problem as an example.

**Sparse Parity Problem.** Given $d$ binary variables $x = (b_1, b_2, \ldots, b_d)$, a sparse parity function selects $k$ secret indices $S = \{i_1, i_2, \ldots, i_k\}$ and outputs 1 if the sum of the corresponding variables is odd, and 0 otherwise:

$$\mathrm{parity}_S(b_1, b_2, \ldots, b_d) = b_{i_1} \oplus b_{i_2} \oplus \cdots \oplus b_{i_k},$$

where $\oplus$ denotes the XOR (exclusive OR) operation. We define $Parity(d, k)$ as the set of all parity functions with $d$ variables and $k$ secret indices, yielding a total of

$$|Parity(d, k)| = \binom{d}{k} = O(d^k).$$

**Representation Matters.** Without Chain-of-Thought (CoT), the sparse parity problem $Parity(d, k)$ is of $ARC(\binom{d}{k}, 1)$, but with CoT, it becomes an autoregressive compositional structure of $ARC(d, k)$.
- No CoT $\rightarrow ARC\left(\binom{d}{k}, 1\right)$.

- With CoT $\rightarrow ARC(d, k)$.

Indeed, without CoT, the model maps input $x = (b_1, \ldots, b_d)$ directly to output $y = b_{i_1} \oplus b_{i_2} \oplus \cdots \oplus b_{i_k}$ in a single step, hence we have length $T = 1$ and the breadth $D = |Parity(d, k)| = O(d^k)$. Such exponential dependency on the breadth suggests that one would need to train on $O(d^k)$ tasks, explaining what we are observing unsuccessful task generalization in the introduction figure.

In contrast, with CoT (Abbe et al., 2024; Wen et al., 2025), the parity computation is decomposed into small steps:

$$\boldsymbol{y} = (b_{i_1}, b_{i_1} \oplus b_{i_2}, \ldots, b_{i_1} \oplus \cdots \oplus b_{i_k}),$$

enabling a structured representation that reduces the breadth. More precisely, at step $t$, the class of subtask is exactly defined by the XOR operation with previous token and one secret index :

$$P_{(t,i_t)}(y_t|\boldsymbol{x}, \boldsymbol{y}_{<t}) = \mathbf{1}[y_t = y_{t-1} \oplus b_{i_t}].$$

where $i_t$ is the $t$-th secret index, by default lying $\in [1, d]$. Therefore, the breadth $D$ at each step is exactly $d$.

This said, the CoT effectively reduces the breadth from $O(d^k)$ to $d$. According to Theorem 3.3,

**Corollary 3.5.** *For the sparse parity problem described above, we can show that the parameter $c$ in Assumption 3.2 is $\frac{1}{2}$, thus when $n_\theta \geq d\ln(100kd), \ell \geq 8\ln(100kn_\theta)$ , it holds that*

$$\lim_{n_x \to \infty} \Pr\left[\mathcal{A}\left(\mathcal{D}_{\text{infer}}; \mathcal{D}_{\text{train}}\right) \neq (P_{(1,i_1)}, \cdots, P_{(k,i_k)})\right] \leq 0.02.$$

In other words, the family of $Parity(d, k)$ with CoT is learnable with $O(d\log(d))$ training tasks.

# 4. Experiments: Parity Problem Case Study

As we have shown, there exists a learning algorithm that is only trained on $\tilde{O}(d)$ different tasks to fully generalize on all the tasks in $Parity(d, k)$. However, from a practical standpoint, it is not clear whether a *Transformer* can actually match this task complexity. In this section, we present empirical evidence demonstrating that a standard Transformer can indeed learn the sparse parity function with CoT using $\tilde{O}(d)$ training tasks. The paper's GitHub repository can be found online.

## 4.1. Experimental Setup: In-Context Learning

Our empirical setup is a refinement of the theoretical framework presented in Section 3.3, and closely follows that of (Garg et al., 2022; Bhattamishra et al., 2024). In this setup, a sequence model $M$ (such as Transformers) is trained using $N$ sequences, each sequence consisting of $m$ demonstration samples $(\boldsymbol{x}_1, \boldsymbol{y}_1, \ldots, \boldsymbol{x}_m, \boldsymbol{y}_m)$. The model is trained for the next token-prediction task, except that we only consider $y_j$ in loss optimization: for each context $(\boldsymbol{x}_1, \boldsymbol{y}_1, \ldots, \boldsymbol{x}_{j-1}, \boldsymbol{y}_{j-1}, \boldsymbol{x}_j)$, the model predicts $\hat{\boldsymbol{y}}_j$, and the loss is given by $\frac{1}{m}\sum_{j=1}^{m}\ell(\hat{\boldsymbol{y}}_j, \boldsymbol{y}_j)$. In our experiment, we use cross-entropy loss to measure the discrepancy between the predicted output $\hat{\boldsymbol{y}}_j$ and the true output $\boldsymbol{y}_j$.

When Chain of Thought (CoT) is used, each $y_j$ is itself a sequence of length $k$ representing intermediate reason-ing steps. In this case, the loss is the average on all these intermediate steps.

**Training Data Generation.** We split both the task space and input space into training and testing. In other words, the parity tasks are split into $\mathcal{F}_{\text{train}}$ and $\mathcal{F}_{\text{test}}$; and the binary sequences are split into $\mathcal{X}_{\text{train}}$ and $\mathcal{X}_{\text{test}}$. The split in the input space helps us monitor the in-distribution training process while as the split in the task space aims to measure generalization to unseen parity functions.

To construct the ICL data, we sample $m$ points $\boldsymbol{x}_1, \ldots, \boldsymbol{x}_m$ uniformly at random from $\mathcal{X}_{\text{train}}$. Similarly, we sample a function $f$ uniformly at random from $\mathcal{F}_{\text{train}}$, and generate the sequence $(\boldsymbol{x}_1, f(\boldsymbol{x}_1), \ldots, \boldsymbol{x}_m, f(\boldsymbol{x}_m))$. This process is repeated $N$ times, each time with a fresh sample of $m$ points and a new function $f$.

**Evaluating Task Generalization.** For evaluation, we sample $f$ randomly from the held-out tasks $\mathcal{F}_{test}$ and sample inputs uniformly at random from $\mathcal{X}_{\text{test}}$. We report the accuracy of the prediction $f(\boldsymbol{x}_m)$ given the demonstration $(\boldsymbol{x}_1, f(\boldsymbol{x}_1), \ldots, \boldsymbol{x}_{m-1}, f(\boldsymbol{x}_{m-1}), \boldsymbol{x}_m)$. This setting challenges the model to generalize to novel tasks beyond those encountered during training.

## 4.2. Experimental Results

As discussed in Section 3.5, introducing CoT transforms the parity problem class $Parity(d, k)$ into an AutoRegressive Compositional structure $ARC(D, T)$ with $D = d$ and $T = k$. A key empirical question is:

*how does the number of training tasks scale with $d$ and $T$ to achieve a fixed target accuracy on unseen task sets?*

To investigate this, we conduct experiments on:

1. Scaling $T(= k)$, i.e. the length of secret indices.

2. Scaling $D(= d)$, i.e. the ambient dimension of input.

**Scaling $T$ for a Fixed $D$.** We examine how the number of training tasks affects test accuracy while keeping the ambient dimension fixed at $d = 15$. Specifically, we evaluate test accuracy for $k = \{3, 4, 5, 6, 7\}$ under varying numbers of training tasks. As $k$ increases, the size of the parity class grows significantly—from approximately 500 for $k = 3$ to around 6500 for $k = 7$.

Remarkably, despite this increase, the test accuracy follows a similar trajectory. With just $3d\ln(d) \approx 122$ training tasks, the model generalizes to unseen cases with high accuracy ($> 95\%$). For $k = 7$, this means training on 122 tasks enables generalization to about 6,400 unseen ones! This empirical results suggests that the required number of training tasks remains roughly the same, regardless of $k$, consistent with the theoretical scaling of $\tilde{O}(d)$ tasks.

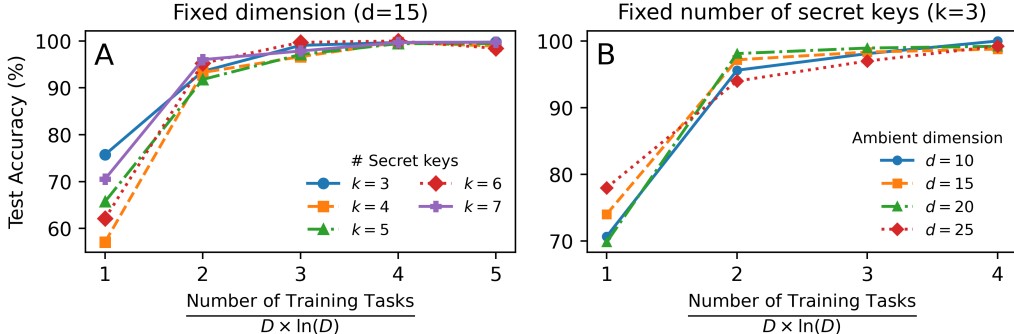

*Figure 3.* Test accuracy on unseen tasks. For parity task: $D = d$ as the ambient dimension and $T = k$ as the number of secret indices. We show that the empirical scaling closely follows the theoretical scaling of $D \ln(D)$. **(A)** For a fixed $D = 15$, as $T$ increases, the test accuracy on unseen tasks remains similar, even though the total number of tasks ($\sim D^T$) grows exponentially with $T$. **(B)** For a fixed secret length is 3, as $D$ increases, the number of tasks grows polynomially with $D$, yet the number of tasks required to generalize reasonably to unseen tasks remains in $\propto D \log D$.

**Scaling $D$ for a fixed $T$.** We examine the effect of increasing $d \in \{10, 15, 20\}$ while keeping $k = 3$ fixed. For each $d$, we train on a total number of tasks proportional to $d \ln(d)$, up to $4 \times d \ln(d)$. Figure 3, Panel B, shows similar task generalization performance across different ambien dimension of $d$, providing further evidence that $\tilde{O}(d)$ i.i.d. training tasks are sufficient for generalization to unseen tasks on praity functions with CoT.

**Scaling $D$ and $T$ together.** We examine the effect of jointly increasing $d$ and $k$, with $(d, k) \in \{(10, 5), (15, 7), (20, 10), (25, 12), (30, 15)\}$. For each $d$, the model is trained on a total of $3 \times d \ln(d)$ i.i.d. tasks. Table 1 shows that generalization performance remains consistent across these settings, providing further evidence that $\tilde{O}(d)$ training tasks are sufficient to generalize to unseen parity tasks using CoT prompting. As a concrete example, for $d = 30, k = 15$, training on just 306 tasks enables generalization to approximately 155 million unseen tasks.

| $d$ | $k$ | #TrainingTasks | #TotalTasks | Accuracy (%) |
|-----|-----|----------------|-------------|--------------|
| 10 | 5 | 69 | 252 | 98.51 |
| 15 | 7 | 121 | 6,400 | 99.12 |
| 20 | 10 | 180 | 185,000 | 98.67 |
| 25 | 12 | 241 | 3,200,000 | 98.60 |
| 30 | 15 | 306 | 155,000,000 | 98.10 |

*Table 1.* Task generalization performance as $d$ and $k$ increase. Training on only $\tilde{O}(d)$ tasks enables generalization to exponentially many unseen tasks in the parity function family.

**Subtask Identification via Linear Probing.** Finally, we probe the hidden representations of the Transformer (see (Alain & Bengio, 2017)) to see if it *identifies and then executes subtasks* at the inference time, consistent with the framework of Section 3. Specifically, we add a linear classifier to the final attention layer's hidden state when producing

the $i$-th token in the Chain-of-Thought, aiming to predict the $i$-th secret index. Only the linear classifier is trained, while all Transformer parameters remain frozen. Table 2 shows that for $d \in \{10, 15, 20\}$, $k = 3$, the linear probe consistently achieves high accuracy at all the coordinates. This suggests that at each CoT step, the model indeed first infers the relevant subtask (the secret index) from the in-context examples and then executes that subtask to generate the output token—an ability it acquires during training. Further experimental details appear in Appendix D.

| $d$ | Token 1 | Token 2 | Token 3 |
|-----|---------|---------|---------|
| 10 | 100.00% | 99.83% | 91.08% |
| 15 | 95.42% | 99.97% | 98.73% |
| 20 | 97.38% | 95.54% | 91.92% |

*Table 2.* Validation accuracy (%) of linear probes trained to predict each secret token position from the final hidden state.

## 5. Task Generalization Beyond i.i.d. Sampling and Parity Functions

In this section, we extend our experiments beyond i.i.d. task sampling and parity functions. We show an adversarial example where biased task selection substantially hinders task generalization for sparse parity problem. In addition, we demonstrate that exponential task scaling extends to a non-parity tasks including arithmetic and multi-step language translation.

### 5.1. Task Generalization Beyond i.i.d. Task Sampling

In previous sections, we focused on *i.i.d. settings*, where the set of training tasks $\mathcal{F}_{train}$ were sampled uniformly at random from the entire class $\mathcal{F}$. Here, we explore scenarios that deliberately break this uniformity to examine the effect of task selection on out-of-distribution (OOD) generalization.

*How does the selection of training tasks influence a model's ability to generalize to unseen tasks? Can we predict which setups are more prone to failure?*

To investigate this, we consider two cases parity problems with $d = 10$ and $k = 3$, where each task is represented by its tuple of secret indices $(s_1, s_2, s_3)$:

1. **Generalization with a Missing Coordinate.** In this setup, we exclude all training tasks where the second coordinate takes the value $s_2 = 5$, such as $(1, 5, 7)$. At test time, we evaluate whether the model can generalize to unseen tasks where $s_2 = 5$ appears.

2. **Generalization with Missing Pair.** Here, we remove all training tasks that contain both $4$ *and* $6$ in the tuple $(s_1, s_2, s_3)$, such as $(2, 4, 6)$ and $(4, 5, 6)$. At test time, we assess whether the model can generalize to tasks where both $4$ and $6$ appear together.

**If you had to guess.** Which scenario is more challenging for generalization to unseen tasks? We provide the experimental result in Table 3.

In the first scenario, despite being trained on all tasks except those where $s_2 = 5$, which is of size $O(D^T)$, the model struggles to generalize to these excluded cases, with prediction at chance level. This is intriguing as one may expect model to generalize across position. The failure suggests that positional diversity plays a crucial role in the task generalization of Transformers.

In contrast, in the second scenario, though the model has never seen tasks with both $4$ *and* $6$ together, it has encountered individual instances where $4$ appears in the second position (e.g., $(1, 4, 5)$) or where $6$ appears in the third position (e.g., $(2, 3, 6)$). This exposure appears to facilitate generalization to test cases where both $4$ *and* $6$ are present. As a result, when the training tasks are not i.i.d, an adversarial selection such as exclusion of specific positional configurations may lead to failure to unseen task generalization even though the size of $\mathcal{F}_{train}$ is exponentially large.

## 5.2. Task Generalization Beyond Parity Problems

### 5.2.1. ARITHMETIC TASK

We introduce the family of *Arithmetic* task that, like the sparse parity problem, operates on $d$ binary inputs $b_1, b_2, \ldots, b_d$. The task involves computing a structured arithmetic expression over these inputs using a sequence of addition and multiplication operations.

Formally, we define the function:

$$\text{Arithmetic}_S \colon \{0, 1\}^d \to \{0, 1, \ldots, d\},$$

where $S = (\text{op}_1, \text{op}_2, \ldots, \text{op}_{d-1})$ is a sequence of $d - 1$ operations, each $\text{op}_k$ chosen from $\{+, \times\}$. The function

evaluates the expression by applying the operations sequentially from left-to-right order: for example, if $S = (+, \times, +)$, then the arithmetic function would compute $\text{Arithmetic}_S(b_1, b_2, b_3, b_4) = ((b_1 + b_2) \times b_3) + b_4$.

By introducing a step-by-step CoT, arithmetic class belongs to $ARC(2, d-1)$: this is because at every step, there is only $D = |\Theta_t| = 2$ choices (either $+$ or $\times$) while the length is $T = d - 1$, resulting a total number of $2^{d-1}$ tasks.

Task generalization for the arithmetic task with CoT. It has $d = 2$ and $T = d - 1$ as the ambient dimension, hence $D \ln(DT) = 2 \ln(2T)$. We show that the empirical scaling closely follows the theoretical scaling.

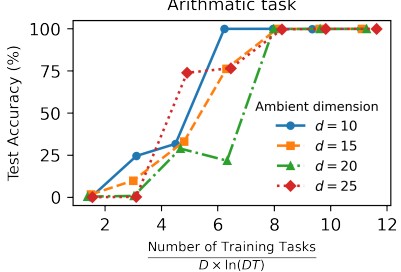

Arithmatic task

Notably, when scaling with $T$, we observe in the figure above that the task scaling closely follow the theoretical $O(D \log(DT))$ dependency. Given that the function class grows exponentially as $2^T$, it is truly remarkable that training on only a few hundred tasks enables generalization to an exponentially larger space—on the order of $2^{25} > 33$ Million tasks. This exponential scaling highlights the efficiency of structured learning, where a modest number of training examples can yield vast generalization capability.

### 5.2.2. MULTI-STEP LANGUAGE TRANSLATION TASK

In this task, we study a sequential translation process across multiple languages (Garg et al., 2022). Given a set of $D$ languages, we construct a translation chain by randomly sampling a sequence of $T$ languages **with replacement**: $L_1, L_2, \ldots, L_T$, where each $L_t$ is a sampled language. Starting with a word, we iteratively translate it through the sequence:

$$L_1 \to L_2 \to L_3 \to \cdots \to L_T.$$

For example, if the sampled sequence is EN $\to$ FR $\to$ DE $\to$ FR, translating the word "butterfly" follows:

butterfly $\to$ papillon $\to$ schmetterling $\to$ papillon.

This task follows an *AutoRegressive Compositional* structure by itself, specifically $ARC(D, T-1)$, where at each step, the conditional generation only depends on the target language, making $D$ as the number of languages and the total number of possible tasks is $D^{T-1}$. This example illustrates that autoregressive compositional structures naturally arise in real-world languages, even without explicit CoT.

We examine task scaling along $D$ (number of languages) and $T$ (sequence length). As shown in Figure 5, empiri-

*Table 3.* Generalization Results for Scenarios 1 and 2 for $d = 10, k = 3$.

| Scenario | Tasks excluded from training | Generalization accuracy |
|---|---|---|
| Generalization with Missing Pair | $\{4, 6\} \subseteq \{s_1, s_2, s_3\}$ | 96.2% |
| Generalization with Missing Coordinate | $s_2 = 5$ | 45.6% |

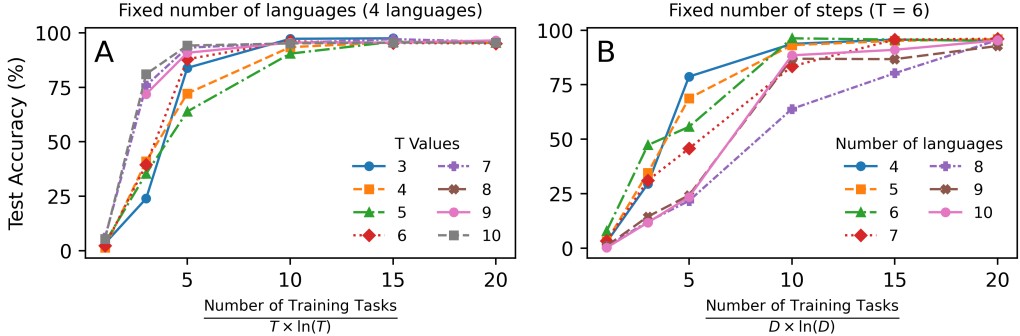

*Figure 5.* Task generalization for language translation task: $D$ is the number of languages and $T$ is the length of steps.

cal $D$-scaling closely follows the theoretical $O(D \ln DT)$. However, in the $T$-scaling case, we observe a linear dependency on $T$ rather than the logarithmic dependency $O(\ln T)$. A possible explanation is error accumulation across sequential steps—longer sequences require higher precision in intermediate steps to maintain accuracy. This contrasts with our theoretical analysis, which focuses on asymptotic scaling and does not explicitly account for compounding errors in finite-sample settings.

Despite this, the task scaling is still remarkable — training on a few hundred tasks enables generalization to $4^{10} \approx 10^6$ tasks!

## 6. Conclusions

In this work, we quantitatively investigated task generalization under the autoregressive compositional structure, demonstrating both theoretically and empirically that exponential task generalization to $D^T$ tasks can be achieved with training on only $\tilde{O}(D)$ tasks. To summarize:

- **Theoretical Framework for Task Generalization.** We introduced the *AutoRegressive Compositional* (ARC) framework to model structured task learning, demonstrating that a model trained on only $\tilde{O}(D)$ tasks can generalize to an exponentially large space of $D^T$ tasks.

- **Formal Sample Complexity Bound.** We established a fundamental scaling law that quantifies the number of tasks required for generalization, proving that exponential generalization is theoretically achievable with only a logarithmic increase in training samples.

- **Empirical Validation on Parity Functions.** We showed that Transformers struggle with standard in-

context learning (ICL) on parity tasks but achieve exponential generalization when Chain-of-Thought (CoT) reasoning is introduced. Our results provide the first empirical demonstration of structured learning enabling efficient generalization in this setting.

- **Scaling Laws in Arithmetic and Language Translation.** Extending beyond parity functions, we demonstrated that the same compositional principles hold for arithmetic operations and multi-step language translation, confirming that structured learning significantly reduces the task complexity required for generalization.

- **Impact of Training Task Selection.** We analyzed how different task selection strategies affect generalization, showing that adversarially chosen training tasks can hinder generalization, while diverse training distributions promote robust learning across unseen tasks.

We introduce a framework for studying the role of compositionality in learning tasks and how this structure can significantly enhance generalization to unseen tasks. Additionally, we provide empirical evidence on learning tasks, such as the parity problem, demonstrating that transformers follow the scaling behavior predicted by our compositionality-based theory. Future research will explore how these principles extend to real-world applications such as program synthesis, mathematical reasoning, and decision-making tasks.

By establishing a principled framework for task generalization, our work advances the understanding of how models can learn efficiently beyond supervised training and adapt to new task distributions. We hope these insights will inspire further research into the mechanisms underlying task generalization and compositional generalization.

## Acknowledgements

We acknowledge support from the National Science Foundation (NSF) and the Simons Foundation for the Collaboration on the Theoretical Foundations of Deep Learning through awards DMS-2031883 and #814639 as well as the TILOS institute (NSF CCF-2112665) and the Office of Naval Research (ONR N000142412631). This work used the programs (1) XSEDE (Extreme science and engineering discovery environment) which is supported by NSF grant numbers ACI-1548562, and (2) ACCESS (Advanced cyberinfrastructure coordination ecosystem: services & support) which is supported by NSF grants numbers #2138259, #2138286, #2138307, #2137603, and #2138296. Specifically, we used the resources from SDSC Expanse GPU compute nodes, and NCSA Delta system, via allocations TG-CIS220009.

## Impact Statement

This paper advances the theoretical understanding of task generalization, investigating how learning on a small subset of tasks enables generalization to a much larger set of unseen tasks. Our study is primarily theoretical, supported by synthetic experiments designed to isolate key structural properties of task composition. By analyzing autoregressive compositional structures, we provide insights into sample complexity and task transferability. While our work lays foundational groundwork for broader applications in machine learning, its immediate impact remains theoretical, with no direct societal implications requiring emphasis.

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

# A. Proofs in Section 3

**Notations** For a distribution over finite class $\mathcal{S}$, denote $\text{supp}(P) \subseteq \mathcal{S}$ as the support of $P$. For two distributions $P, Q$ over finite set $\mathcal{S}$, denote the cross entropy as

$$H(P, Q) = -\sum_{x \in \mathcal{S}} P(x) \log Q(x),$$

and denote the KL divergence as

$$KL(P, Q) = -\sum_{x \in \mathcal{S}} P(x) \log \frac{Q(x)}{P(x)}.$$

## A.1. Proof of Theorem 3.3

To identify the true distribution $\tilde{\theta}$ from $\Theta$ at inference time, we need the following lemma to provide a non-asymptotic bound for distribution discrimination.

**Lemma A.1** (Non-Asymptotic Discrimination of Two Distributions). *Let $\mathcal{S}$ be any finite set and $P, Q \in \Delta(\mathcal{S})$ be two distributions with total variation distance $\text{TV}(P, Q) = c > 0$. Suppose we observe $n$ independent samples $X_1, \ldots, X_n$ from an unknown distribution $Y$, where $Y$ is either $P$ or $Q$. Then, there exists a testing algorithm that identifies $Y$ with probability at least $1 - \delta$ provided that*

$$n \geq \frac{2 \ln(1/\delta)}{c^2}.$$

*Proof.* **Testing Procedure.** Define the testing statistic $\phi$ as follows:

$$\phi = \frac{1}{n} \sum_{i=1}^{n} (-1)^{\mathbf{1}[P(X_i) < Q(X_i)]}.$$

Equivalently, $\phi$ can be written in terms of the empirical distribution $\hat{Y}_n$:

$$\phi = \sum_{x \in \mathcal{S}} \hat{Y}_n(x) \cdot (-1)^{\mathbf{1}[P(x) < Q(x)]},$$

where $\hat{Y}_n(x) = \frac{1}{n} \sum_{i=1}^{n} \mathbf{1}[X_i = x]$.

Compute the expected values of $\phi$ under $Y = P$ and $Y = Q$:

$$\mu_P = \sum_{x \in \mathcal{S}} P(x) \cdot (-1)^{\mathbf{1}[P(x) < Q(x)]}, \quad \mu_Q = \sum_{x \in \mathcal{S}} Q(x) \cdot (-1)^{\mathbf{1}[P(x) < Q(x)]}.$$

The algorithm reports $\hat{Y} = P$ if $|\phi - \mu_P| < |\phi - \mu_Q|$, and $\hat{Y} = Q$ otherwise.

**Proof of Correctness.** Without loss of generality, assume $Y = P$. We analyze the probability of error:

$$\Pr\left(\hat{Y} \neq P\right) = \Pr\left(|\phi - \mu_P| \geq |\phi - \mu_Q|\right).$$

Note that $\mathbb{E}[\phi] = \mu_P$ under $Y = P$. From the definition of total variation distance:

$$\mu_Q - \mu_P = \sum_{x \in \mathcal{S}} (Q(x) - P(x)) \cdot (-1)^{\mathbf{1}[P(x) < Q(x)]} = 2 \cdot d_{\text{TV}}(P, Q) = 2c.$$

Thus, the error probability can be bounded as:

$$\Pr\left(|\phi - \mu_P| \geq |\phi - \mu_Q|\right) \leq \Pr\left(\phi \geq \mu_P + c\right).$$

Since $\phi$ is the average of $n$ independent random variables taking values in $\{-1, +1\}$, by Hoeffding's inequality:

$$\Pr\left(\phi \geq \mu_P + c\right) \leq \exp\left(-\frac{c^2 n}{2}\right).$$

Setting $\exp\left(-\frac{c^2 n}{2}\right) \leq \delta$ gives the required sample complexity $n \geq \frac{2 \ln(1/\delta)}{c^2}$. $\qquad\square$

To prove Theorem 3.3, we introduce the following lemma:

**Lemma A.2.** *Consider a compositional task class $\mathcal{F}$ satisfying Assumption 3.2. Suppose the training tasks $\mathcal{F}_{\text{train}} = \{f_{\theta^i}\}_{i=1}^{n_\theta}$ satisfy the **per-component coverage** condition: for every timestep $t \in [T]$,*

$$\{P_{\theta_t^i}\}_{i=1}^{n_\theta} = \mathcal{P}_{\Theta_t}.$$

*Then there exists a learner $\mathcal{A}$ such that when trained on $\mathcal{D}_{\text{train}} = \{\mathcal{D}_i\}_{i=1}^{n_\theta}$ with $\mathcal{D}_i \overset{\text{i.i.d.}}{\sim} P_{\theta^i}$ where $\mathcal{D}_i$ consists of $n_x$ i.i.d. samples, and given $\ell \geq \frac{2\ln(100 T n_\theta)}{c^2}$ inference-time demonstration samples i.i.d. sampled from unseen task $P_{\tilde{\theta}}$, denoted by $\mathcal{D}_{\text{infer}}$, then it holds that:*

$$\lim_{n_x \to \infty} \Pr\left[\mathcal{A}(\mathcal{D}_{\text{infer}}; \mathcal{D}_{\text{train}}) \neq (P_{\tilde{\theta}_1}, \ldots, P_{\tilde{\theta}_T})\right] \leq 0.01,$$

*where the probability is over the randomness in $\mathcal{D}_{\text{train}}$ and $\mathcal{D}_{\text{infer}}$.*

*Proof.*

**Step 1: Training Stage.**

We construct a learning algorithm that recovers

$$f_{\theta^i} = \left(P_{\theta_1^i}, \ldots, P_{\theta_T^i}\right)$$

from

$$\mathcal{D}_i = \{(\boldsymbol{x}^{i,j}, \boldsymbol{y}^{i,j})\}_{j=1}^{n_x} \overset{\text{i.i.d.}}{\sim} P_{\theta^i}(\boldsymbol{x}, \boldsymbol{y}),$$

which is part of the training set $\mathcal{D}_{\text{train}}$. The procedure is shown below:

---
**Algorithm 1** Training Stage

---
**Require:** Training set $\mathcal{D}_{\text{train}} = \{\mathcal{D}_i\}_{i=1}^{n_\theta}$
1: **for** $i = 1$ to $n_\theta$ **do**
2:     **for** $t = 1$ to $T$ **do**
3:         $\text{Observed\_Support}_t \leftarrow \{(\boldsymbol{x}^{i,j}, \boldsymbol{y}_{1:t}^{i,j})\}_{j=1}^{n_x}$
4:         $\mathcal{P}_{\Xi_t'} \leftarrow \left\{P_{\xi_t} \in \mathcal{P}_{\Xi_t} : \text{supp}(P_{\theta_{1:t-1}^i, \xi_t}) = \text{Observed\_Support}_t\right\}$
5:         $\hat{\theta}_t^i \leftarrow \arg\max_{\xi_t \in \Xi_t'} \sum_{j=1}^{n_x} \log P_{\hat{\theta}_{1:t-1}^i, \xi_t}(\boldsymbol{x}^{i,j}, \boldsymbol{y}_{1:t}^{i,j})$
6:     **end for**
7: **end for**
8: **return** $\mathcal{P}_{\hat{\Theta}_t} = \{P_{\hat{\theta}_t^i}\}_{i=1}^{n_\theta}$ for each $t \in [T]$.

---

We now show that, as $n_x \to \infty$,

$$\Pr\left[\mathcal{P}_{\hat{\Theta}_t} = \mathcal{P}_{\Theta_t} \;\; \forall t \in [T]\right] \to 1.$$

Assume $\hat{\theta}_{1:t-1}^i = \theta_{1:t-1}^i$. As $n_x \to \infty$,

$$\Pr\left[\text{Observed\_Support}_t \neq \text{supp}(P_{\theta_{1:t}^i})\right] \leq \sum_{(\boldsymbol{x}, \boldsymbol{y}_{1:t}) \in \text{supp}(P_{\theta_{1:t}^i})} \Pr\left[(\boldsymbol{x}^{i,j}, \boldsymbol{y}_{1:t}^{i,j}) \neq (\boldsymbol{x}, \boldsymbol{y}_{1:t}) \;\; \forall j \in [n_x]\right] \to 0.$$

Conditioned on $\text{Observed\_Support}_t = \text{supp}(P_{\theta_{1:t}^i})$, any $P_{\xi_t}$ in $\mathcal{P}_{\Xi_t'}$ must share the same support:

$$\text{supp}(P_{\theta_{1:t-1}^i, \xi_t}) = \text{supp}(P_{\theta_{1:t-1}^i, \theta_t^i}).$$

Thus the cross-entropy is finite. By the law of large numbers and Pinsker's inequality, for $P_{\xi_t} \neq P_{\theta_t^i}$ we have

$$\lim_{n_x \to \infty} \left( \frac{1}{n_x} \sum_{j=1}^{n_x} \log P_{\theta^i_{1:t-1}, \theta^i_t}(\boldsymbol{x}^{i,j}, \boldsymbol{y}^{i,j}_{1:t}) - \frac{1}{n_x} \sum_{j=1}^{n_x} \log P_{\theta^i_{1:t-1}, \xi_t}(\boldsymbol{x}^{i,j}, \boldsymbol{y}^{i,j}_{1:t}) \right)$$

$$\to KL(P_{\theta^i_{1:t-1}, \theta^i_t}, P_{\theta^i_{1:t-1}, \xi_t})$$

$$\geq 2\text{TV}(P_{\theta^i_{1:t-1}, \theta^i_t}, P_{\theta^i_{1:t-1}, \xi_t})^2$$

$$> 0$$

almost surely. Thus,

$$\Pr\left[P_{\hat{\theta}^i_t} \neq P_{\theta^i_t}\right] \leq \sum_{\xi_t \in \Xi'_t, \, P_{\xi_t} \neq P_{\theta^i_t}} \Pr\left[\frac{1}{n_x} \sum_{j=1}^{n_x} \log P_{\theta^i_{1:t-1}, \xi_t}(\boldsymbol{x}^{i,j}, \boldsymbol{y}^{i,j}_{1:t}) \geq \frac{1}{n_x} \sum_{j=1}^{n_x} \log P_{\theta^i_{1:t-1}, \theta^i_t}(\boldsymbol{x}^{i,j}, \boldsymbol{y}^{i,j}_{1:t})\right] \to 0.$$

Recursively applying this argument from $t = 1$ to $T$ and taking a union bound over all $i \in [n_\theta]$ and $t \in [T]$,

$$\lim_{n_x \to \infty} \Pr\left[\exists (t, i) : P_{\hat{\theta}^i_t} \neq P_{\theta^i_t}\right] = 0.$$

Finally, by the assumption that $\{P_{\theta^1_t}, \ldots, P_{\theta^{n_\theta}_t}\} = \mathcal{P}_{\Theta_t}$ for each $t$, we conclude

$$\Pr\left[\mathcal{P}_{\hat{\Theta}_t} = \mathcal{P}_{\Theta_t} \; \forall t \in [T]\right] \to 1 \quad \text{as} \quad n_x \to \infty.$$

**Step 2: Inference Stage**

Assume that, in the training phase, the learner identifies the true distribution sets

$$\mathcal{P}_{\hat{\Theta}_t} = \{P_{\hat{\theta}^1_t}, \ldots, P_{\hat{\theta}^{n_\theta}_t}\} = \mathcal{P}_{\Theta_t} \quad \text{for all } t = 1, \ldots, T.$$

We now construct an algorithm that, given $\ell \geq \frac{2 \ln\left(100 \, T \, n_\theta\right)}{c^2}$ independent samples from the unseen composite task

$$f_{\tilde{\theta}} = \left(P_{\hat{\theta}_1}, \ldots, P_{\hat{\theta}_T}\right),$$

outputs the same distribution tuple $\left(P_{\tilde{\theta}_1}, \ldots, P_{\tilde{\theta}_T}\right)$ with probability at least $0.99$.

To achieve this, we leverage the distribution discrimination technique of Lemma A.1 to distinguish between candidates in $\mathcal{P}_{\hat{\Theta}_t}$, given sufficiently many demonstration samples.

---

**Algorithm 2** Inference Stage

---

**Require:** Inference-time demonstration samples $\mathcal{D}_{\text{infer}} = \{(\tilde{\boldsymbol{x}}^j, \tilde{\boldsymbol{y}}^j)\}_{j=1}^\ell$ i.i.d. from $P_{\tilde{\theta}}$, and the identified sets $\mathcal{P}_{\hat{\Theta}_t} = \{P_{\hat{\theta}_t^1}, \ldots, P_{\hat{\theta}_t^{n_\theta}}\}$ for each $t \in [T]$.

1: **for** $t = 1$ to $T$ **do**
2:    Initialize $P_{\zeta_t} \leftarrow P_{\hat{\theta}_t^1}$.
3:    **for** $i = 2$ to $n_\theta$ **do**
4:       Compute

$$\phi \leftarrow \frac{1}{\ell} \sum_{j=1}^\ell (-1)^{\mathbf{1}\left[P_{\zeta_{1:t-1}, \zeta_t}(\tilde{\boldsymbol{x}}^j, \tilde{\boldsymbol{y}}_{1:t}^j) < P_{\zeta_{1:t-1}, \hat{\theta}_t^i}(\tilde{\boldsymbol{x}}^j, \tilde{\boldsymbol{y}}_{1:t}^j)\right]}.$$

5:       **if**

$$\left| \phi - \sum_{(\boldsymbol{x}, \boldsymbol{y}_{1:t}) \in \mathcal{X} \times \mathcal{Y}^t} P_{\zeta_{1:t-1}, \hat{\theta}_t^i}(\boldsymbol{x}, \boldsymbol{y}_{1:t}) (-1)^{\mathbf{1}\left[P_{\zeta_{1:t-1}, \zeta_t}(\boldsymbol{x}, \boldsymbol{y}_{1:t}) < P_{\zeta_{1:t-1}, \hat{\theta}_t^i}(\boldsymbol{x}, \boldsymbol{y}_{1:t})\right]} \right|$$

$$< \left| \phi - \sum_{(\boldsymbol{x}, \boldsymbol{y}_{1:t}) \in \mathcal{X} \times \mathcal{Y}^t} P_{\zeta_{1:t-1}, \zeta_t}(\boldsymbol{x}, \boldsymbol{y}_{1:t}) (-1)^{\mathbf{1}\left[P_{\zeta_{1:t-1}, \zeta_t}(\boldsymbol{x}, \boldsymbol{y}_{1:t}) < P_{\zeta_{1:t-1}, \hat{\theta}_t^i}(\boldsymbol{x}, \boldsymbol{y}_{1:t})\right]} \right|.$$

      **then**
6:         Update $P_{\zeta_t} \leftarrow P_{\hat{\theta}_t^i}$.
7:       **end if**
8:    **end for**
9: **end for**
10: **return** $(P_{\zeta_1}, \ldots, P_{\zeta_T})$.

---

**Error Analysis.** Let $P_{\zeta_t}$ be the chosen distribution at step $t$. Given $P_{\zeta_{1:t-1}} = P_{\tilde{\theta}_{1:t-1}}$, Lemma A.1 ensures that any incorrect candidate can be detected with high probability, as long as $P_{\tilde{\theta}_t} \in \mathcal{P}_{\hat{\Theta}_t}$. Specifically,

$$\Pr\left[ P_{\zeta_t} \neq P_{\tilde{\theta}_t} \mid P_{\zeta_{1:t-1}} = P_{\tilde{\theta}_{1:t-1}} \right]$$

$$\leq \sum_{i=2}^{n_\theta} \Pr\left[ \text{distribution testing fails at step } i \mid \zeta_t = \tilde{\theta}_t \text{ or } \hat{\theta}_t^i = \tilde{\theta}_t \right] \leq n_\theta \exp\left(-\frac{c^2 \ell}{2}\right).$$

A union bound over $t = 1, \ldots, T$ then implies

$$\Pr\left[ (P_{\zeta_1}, \ldots, P_{\zeta_T}) \neq (P_{\tilde{\theta}_1}, \ldots, P_{\tilde{\theta}_T}) \right] \leq T n_\theta \exp\left(-\frac{c^2 \ell}{2}\right).$$

Since $\ell \geq \frac{2 \ln\left(100 \, T \, n_\theta\right)}{c^2}$, we get

$$T n_\theta \exp\left(-\frac{c^2 \ell}{2}\right) \leq 0.01.$$

**Step 3: Combining Training and Inference**

From the results of Step 1 and Step 2, we have

$$\lim_{n_x \to \infty} \Pr\left[ \mathcal{A}(\mathcal{D}_{\text{infer}}; \mathcal{D}_{\text{train}}) \neq (P_{\tilde{\theta}_1}, \ldots, P_{\tilde{\theta}_T}) \right]$$

$$\leq \lim_{n_x \to \infty} \left( \Pr\left[ (P_{\zeta_1}, \ldots, P_{\zeta_T}) \neq (P_{\tilde{\theta}_1}, \ldots, P_{\tilde{\theta}_T}) \mid \mathcal{P}_{\hat{\Theta}_t} = \mathcal{P}_{\Theta_t} \; \forall t \right] + \Pr\left[ \exists t \in [T] : \mathcal{P}_{\hat{\Theta}_t} \neq \mathcal{P}_{\Theta_t} \right] \right)$$

$$= 0.01 + 0 = 0.01.$$

$\square$

*Proof of Theorem 3.3.* By Lemma A.2, it suffices to verify that per-component coverage condition holds for each timestep with high probability when $n_\theta = D \ln(100\, D\, T)$.

For each timestep $t \in [T]$, observe

$$\Pr\left[\{P_{\theta_t^i}\}_{i=1}^{n_\theta} \neq \mathcal{P}_{\Theta_t}\right] \leq \sum_{P_{\theta_t} \in \mathcal{P}_{\Theta_t}} \Pr\left[P_{\theta_t^i} \neq P_{\theta_t} \ \forall i \in [n_\theta]\right]$$

$$= D\left(1 - \frac{1}{D}\right)^{n_\theta}.$$

Hence,

$$\Pr\left[\exists\, t \in [T] : \{P_{\theta_t^i}\}_{i=1}^{n_\theta} \neq \mathcal{P}_{\Theta_t}\right] \leq \sum_{t=1}^{T} \Pr\left[\{P_{\theta_t^i}\}_{i=1}^{n_\theta} \neq \mathcal{P}_{\Theta_t}\right]$$

$$\leq D\,T\left(1 - \frac{1}{D}\right)^{n_\theta} < D\,T\,e^{-n_\theta/D} = 0.01.$$

This completes the proof. $\qquad\qquad\qquad\qquad\qquad\qquad\qquad\qquad\qquad\qquad\qquad\qquad\qquad\qquad\qquad\square$

### A.2. Proof of Corollary 3.5

*Proof of Corollary 3.5.* It suffices to verify that for the sparse parity problem, the constant $c$ in Assumption 3.2 is $\frac{1}{2}$.

We denote $P_{i_1, \cdots, i_t}(\boldsymbol{x}, \boldsymbol{y}_{<t}) = \frac{1}{2^d} \times \mathbf{1}[y_1 = x_{i_1}] \prod_{s=2}^{t} \mathbf{1}[y_s = y_{s-1} \oplus x_{i_s}]$. For any $i_1, \cdots, i_{t-1} \in [d]$, and $i_t \neq i_t' \in [d]$ and it holds that

$$\mathrm{TV}(P_{i_1, \cdots, i_{t-1}, i_t}, P_{i_1, \cdots, i_{t-1}, i_t'}) = \frac{1}{2^n} \sum_{x \in \{0,1\}^d} \mathbf{1}[x_{i_1} \oplus \cdots \oplus x_{i_{t-1}} \oplus x_{i_t} \neq x_{i_1} \oplus \cdots \oplus x_{i_{t-1}} \oplus x_{i_t'}]$$

$$= \frac{1}{2^n} \sum_{x \in \{0,1\}^d} \mathbf{1}[x_{i_t} \neq x_{i_t'}]$$

$$= \frac{1}{2}.$$

$\qquad\qquad\qquad\qquad\qquad\qquad\qquad\qquad\qquad\qquad\qquad\qquad\qquad\qquad\qquad\qquad\qquad\qquad\qquad\qquad\qquad\square$

This completes the proof.

## B. Non-Asymptotic Analysis of Training-Time Demonstration Sample Complexity

In Theorem 3.3, we established only an *asymptotic* result, showing that as the number of demonstration samples per task at training time $n_x \to \infty$, the probability of correctly identifying subtask families $\mathcal{P}_{\Theta_t}$ tends to one. However, by imposing an additional assumption on the total variation gap between the true distributions and any other hypotheses, it is possible to derive a *non-asymptotic* guarantee on how large $n_x$ must be for accurate subtask identification.

Although maximum-likelihood estimation (MLE) does not directly yield such a non-asymptotic bound in this setting, we can use the same distribution discrimination approach introduced in the inference stage (Lemma A.1).

**Assumption B.1** (Compositional Identifiability with fixed tv margin)**.** The autoregressive task class $\mathcal{F}$ satisfies:

1. **Finite Subtask Families**: For each $t \in [T]$, the hypothesis class $\mathcal{P}_{\Xi_t}$ is of size at most $H$ and the subtask conditional distribution family $\mathcal{P}_{\Theta_t} \subseteq \mathcal{P}_{\Xi_t}$ has size $|\mathcal{P}_{\Theta_t}| = D$.

2. **Task Identifiability**: For any $t \in [T]$, $\theta_{1:t-1} \in \bigtimes_{s=1}^{t-1} \Theta_s$, and $\theta_t \in \Theta_t$, $\zeta_t \in \Xi_t$, $P_{\zeta_t} \neq P_{\theta_t}$, the induced distributions stasify:
$$\mathrm{TV}\left(P_{\theta_{1:t-1}, \theta_t}, P_{\theta_{1:t-1}, \zeta_t}\right) \geq r > 0.$$

   Furthermore, for any timestep $t \in [T]$, $\theta_{1:t-1} \in \bigtimes_{s=1}^{t-1} \Theta_s$, and $\theta_t \neq \theta_t' \in \Theta_t$, the induced distributions satisfy:
$$\mathrm{TV}\left(P_{\theta_{1:t-1}, \theta_t}, P_{\theta_{1:t-1}, \theta_t'}\right) \geq c > 0.$$

**Theorem B.2** (Exponential Task Generalization). *Let $\mathcal{F}$ be an autoregressive compositional task class satisfying Assumption 3.2. Then there exists a learner $\mathcal{A}$ with the following property: if during training, one samples $n_\theta$ tasks uniformly and independently from $\mathcal{F}$, each provided with $n_x$ i.i.d. demonstration samples as the training dataset, and if at inference one observes $\ell$ i.i.d. demonstration samples from a previously unseen task $P_{\tilde{\theta}} \in \mathcal{F}$, then*

$$\Pr\Big[\mathcal{A}\big(\mathcal{D}_{\text{infer}};\mathcal{D}_{\text{train}}\big) \neq \big(P_{\hat{\theta}_1},\ldots,P_{\hat{\theta}_T}\big)\Big] \leq DTe^{-n_\theta/D} + n_\theta T e^{-c^2\ell/2} + n_\theta T H e^{-r^2 n_x/2}.$$

*where $\mathcal{D}_{\text{train}}$ and $\mathcal{D}_{\text{infer}}$ denote the training dataset and inference-time demonstration samples respectively, and the probability is taken over the random selection of training tasks $\mathcal{F}_{\text{train}} \subseteq \mathcal{F}$, the training data $\mathcal{D}_{\text{train}}$, and the inference time demonstration samples $\mathcal{D}_{\text{infer}}$.*

*Proof.* Denote the hypothesis class $\mathcal{P}_{\Xi_t} = \{P_{\xi_{t,1}},\cdots,P_{\xi_{t,|\Xi_t|}}\}$, we present the training stage of the learner.

---

**Algorithm 3** Training Stage with Distribution Dislimination

---

**Require:** Training set $\mathcal{D}_{\text{train}} = \{\mathcal{D}_i\}_{i=1}^{n_\theta}$
1: **for** $i = 1$ to $n_\theta$ **do**
2:     **for** $t = 1$ to $T$ **do**
3:         Initialize $P_{\hat{\theta}_t^i} \leftarrow P_{\xi_{t,1}}$.
4:         **for** $k = 2$ to $|\Xi_t|$ **do**
5:             Compute

$$\phi \leftarrow \frac{1}{n_x} \sum_{(\boldsymbol{x}^{i,j},\boldsymbol{y}^{i,j})\in\mathcal{D}_i} (-1)^{\mathbf{1}\left[P_{\hat{\theta}_{1:t-1},\hat{\theta}_t^i}(\boldsymbol{x}^{i,j},\boldsymbol{y}_{1:t}^{i,j}) < P_{\hat{\theta}_{1:t-1},\xi_{t,k}}(\boldsymbol{x}^{i,j},\boldsymbol{y}_{1:t}^{i,j})\right]}.$$

6:             **if**

$$\left| \phi - \sum_{(\boldsymbol{x},\boldsymbol{y}_{1:t})\in\mathcal{X}\times\mathcal{Y}^t} P_{\hat{\theta}_{1:t-1},\xi_{t,k}}(\boldsymbol{x},\boldsymbol{y}_{1:t})(-1)^{\mathbf{1}\left[P_{\hat{\theta}_{1:t-1},\hat{\theta}_t^i}(\boldsymbol{x},\boldsymbol{y}_{1:t}) < P_{\hat{\theta}_{1:t-1},\xi_{t,k}}(\boldsymbol{x},\boldsymbol{y}_{1:t})\right]} \right|$$
$$< \left| \phi - \sum_{(\boldsymbol{x},\boldsymbol{y}_{1:t})\in\mathcal{X}\times\mathcal{Y}^t} P_{\hat{\theta}_{1:t-1},\hat{\theta}_t^i}(\boldsymbol{x},\boldsymbol{y}_{1:t})(-1)^{\mathbf{1}\left[P_{\hat{\theta}_{1:t-1},\hat{\theta}_t^i}(\boldsymbol{x},\boldsymbol{y}_{1:t}) < P_{\hat{\theta}_{1:t-1},\xi_{t,k}}(\boldsymbol{x},\boldsymbol{y}_{1:t})\right]} \right|.$$

            **then**
7:                Update $P_{\hat{\theta}_t^i} \leftarrow P_{\xi_{t,k}}$.
8:             **end if**
9:         **end for**
10:     **end for**
11: **end for**
12: **return** $\mathcal{P}_{\hat{\Theta}_t} = \{P_{\hat{\theta}_t^i}\}_{i=1}^{n_\theta}$ for each $t \in [T]$.

---

Using the same approach as in Step 2 of the proof of Theorem 3.3,

$$\Pr[(P_{\hat{\theta}_1^i},\cdots,P_{\hat{\theta}_T^i}) \neq (P_{\theta_1^i},\cdots,P_{\theta_T^i})] \leq THe^{-r^2 n_x/2}.$$

By union bound,

$$\Pr[\exists\, t \in [T] : \mathcal{P}_{\hat{\Theta}_t} \neq \mathcal{P}_{\Theta_t}] \leq \Pr[\exists(t,i) : P_{\hat{\theta}_t^i} \neq P_{\theta_t^i}] \leq n_\theta T H e^{-r^2 n_x/2}.$$

The remainder of the proof then proceeds exactly as in Theorem 3.3.

$\square$

## C. Extra experiments

**Effect of Context Length.** The theory assumes access to an infinite number of examples for each training task but does not require infinite demonstrations during inference. However, in practice, we cannot train on an infinite number of examples. Figure 6 shows that providing sufficient context length during both training and inference is crucial for strong performance. Empirically, we observed that a context length of 40 works reasonably well across all experiments with dimensions up to $d = 20$.

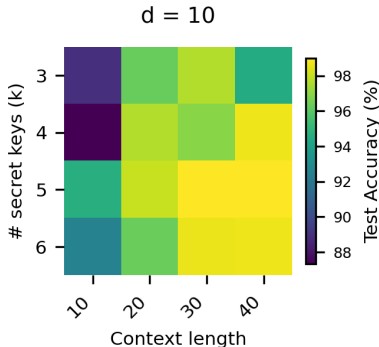

*Figure 6.* The effect of context length on performance.

**ICL with no CoT fails in even in-distribution generalziation.** We observe in Figure 7 that transformers with ICL and no CoT struggle to generalize even in simpler in-distribution settings as the number of tasks increases. In the parity task, we refer to in-distribution generalization as a setting where the model is trained on $\mathcal{F}_{train}$ tasks and $\mathcal{S}_{train}$ sequences, and then evaluated on the same set of tasks $\mathcal{F}_{train}$ but with entirely new sequences $\mathcal{S}_{test}$ that were not seen during training.

Here, the setting is the same as in (Bhattamishra et al., 2024) for Parity(10, 2), but we used the same tasks during both training and testing. We trained on half of the total sequences, $2^9$ and tested on unseen sequences while keeping the tasks unchanged.

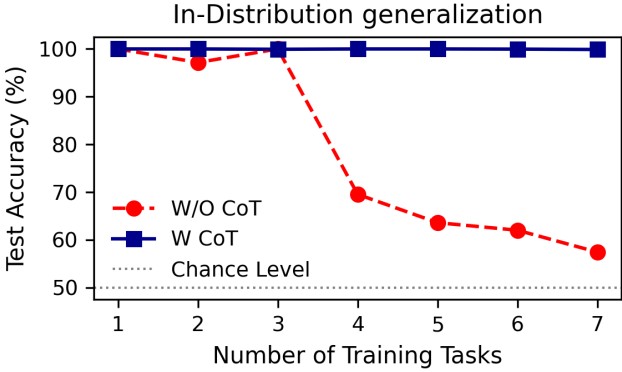

*Figure 7.* ICL without CoT even fails to generalize in distribution.

## D. Experiment Details

**Model and optimization.** We used the transformers library from Hugging Face (Wolf et al., 2020) to instantiate and train our GPT-2 model from scratch. In all experiments, we used a 3-layer, 1-head configuration. We used the Wadam optimizer (Kingma & Ba, 2015) with a learning rate of $8 \times 10^{-5}$ and a batch size of 64.

**Parity and arithmetic.**    In all experiments shown in Figures 3 and 4 for both parity and arithmetic tasks, we used a context length of 40.

For the arithmetic problem, across all dimensions, we used a total of 25,000 training examples, equally distributed across the training tasks.

For the parity problem, we used 20,000 training samples, equally distributed across the training tasks for dimensions up to 15. For dimension 20, we increased the total number of training samples to 50,000.

At testing time, we always randomly select the minimum between 200 subsets and all remaining tasks, each containing 500 different sequences with the same context length of 40.

**Language experiments.**    For the translation experiments, we train a 2-layer Transformer with 3 heads and embedding dimension 768. We use an Adam optimizer with betas being $0.9, 0.95$ and learning rate 3e-4. We will keep the number of total training samples to be $1e5$ and train for 1 pass for 6250 steps. We choose the languages randomly from the following set $\{English, French, Spanish, Chinese, German, Italian, Japanese, Russian,$ $Portuguese, Arabic\}$ and meanings (in English) from $\{cat, dog, house, apple, sky, car, road$ $, tree, bed, water, sun, moon\}$. We use a GPT-2 tokenizer and in our demonstrations, we will prepend the language of the corresponding word before each word in the following format like "English: cat".

**Linear Probing**    We append a linear classifier to the checkpoints of models of "Increasing $D$ for a fixed $T$" tasks, trained on the hidden states of the final attention layer when generating the $i$-th token in the Chain-of-Thought, with the goal of predicting the $i$-th "secret index." The models are trained on a total number of of $20,000, 20,000$, and $50,000$ training samples for $d = 10, 15$, and $20$, respectively. The tasks used for training and validation are disjoint. Only the linear classifier is trained, while the parameters of the transformer are frozen. We use the Adam optimizer with a learning rate of $4 \times 10^{-5}$, and the batch size is set to be 32.

