# OpenReview forum: "Task Generalization with Autoregressive Compositional Structure: Can Learning from $D$ Tasks Generalize to $D^T$ Tasks?"
_ICML.cc/2025/Conference — ICML 2025 poster_

### Official Review · Reviewer_w4XA · 2025-02-27

**Overall Recommendation:** 3

**Summary:**

This paper demonstrates that for Boolean tasks with an AutoRegressive Compositional Structure (e.g., sparse parity), using chain-of-thought (CoT) to break down the tasks into simpler sub-problems—while predicting the outcomes of intermediate steps—significantly improves generalization in the GPT-2 model.

**Claims And Evidence:**

**Weakness 1**

I believe the paper’s most significant contribution is demonstrating that by leveraging chain-of-thought (CoT) to train tasks with an auto-regressive compositional structure:

(1) Tasks that were previously unlearnable without CoT (in the sense that the training loss could not be reduced) can now be learned.
(2) The method also appears to have an advantage in terms of generalization (i.e., the sense that the test loss is small).

However, regarding point (1), I am not particularly surprised, because it seems the authors have used prior knowledge of the ARC structure to explicitly break the tasks down into smaller units via CoT, thereby artificially lowering the complexity and making it more learnable.

As for point (2), I am skeptical of the claim that this approach truly confers a generalization advantage. The paper’s primary example, Sparse Parity, is well-known to be essentially unlearnable (and certainly not generalizable) by Transformers. Thus, from my perspective, the experiments mainly show that CoT makes learning possible in the first place, but do not convincingly demonstrate how much of a generalization benefit is gained over a No CoT approach.

For a fair comparison of generalization performance, I believe it is necessary to examine tasks that can be learned both with and without CoT. Without such a baseline, it seems to me that the paper merely shows how reducing task complexity via CoT enables successful training, which, while potentially useful, is not particularly surprising.

**Essential References Not Discussed:**

The author mentions that, so far, Transformers have been unable to learn sparse parity. However, while reviewing recent studies, this paper* claiming that sparse parity can be learned when multiple tasks are trained together.

*Task Diversity Shortens the ICL Plateau, arXiv 2025. (https://arxiv.org/abs/2410.05448)

**Experimental Designs Or Analyses:**

**Weakness 2**

Despite the fact that the experiments are neither time-consuming nor require extensive GPU resources, the authors did not provide results from multiple repeated trials. As a result, it is difficult to assess the consistency of the findings.

**Methods And Evaluation Criteria:**

I believe the methods and evaluation criteria proposed by the authors are reasonable.

**Other Comments Or Suggestions:**

The multilingual translation experiments are conducted in a relatively simplified setting, where a single word (e.g., "cat" or "dog") is translated through a language chain using a 2-layer, 3-head Transformer. This simplified experimental environment may not adequately capture the diverse linguistic and contextual factors present in real-world translation scenarios, making it difficult to substantiate the claim that the ARC structure naturally arises in actual language translation tasks. Therefore, there are no experimental results that sufficiently bridge the gap between the paper's contributions and real-world scenarios.

**Other Strengths And Weaknesses:**

**Strength**

1. Providing experimental and theoretical explanations of how Transformers can effectively learn and generalize by leveraging CoT.
2. In the sparse parity and arithmetic tasks, it is interesting that the results closely align with the theoretical findings.
3. Overall writing and structure were easy to read!

**Weakness**

Please see weakness 1 and weakness 2.

**Questions For Authors:**

1. How is the input prompt structured when training with CoT? I'm curious about how zero padding is applied. For example, when using 10101 -> 111 as input, do you add two zero paddings to 111, making it 11100? It would be helpful if you could provide a detailed explanation of the overall process. (I understand the setting used in the paper "Understanding In-Context Learning in Transformers and LLMs by Learning to Learn Discrete Functions")

2. I’m curious whether the experimental results are permutation-invariant. There are multiple ways to apply CoT (technically, there are k! possible ways). In the current paper, the model processes the input autoregressively in order, starting from the smallest index (such as figure in page 5). What happens if a randomly permuted order is used instead of ascending order? If you have experimented with this, I’d love to know the results. Additionally, I’d be interested in hearing the authors' insights on this matter.

**Relation To Broader Scientific Literature:**

This work appears to contribute to the research area of using synthetic tasks to better understand language models. The parity task is known to be difficult for Transformers to learn, but other researchers could gain insights from this paper on how to make it learnable.

**Theoretical Claims:**

Under the stated assumptions, the proofs for the theoretical claims are logically coherent and employ standard techniques appropriately. While some assumptions (e.g., infinite samples, fixed TV gaps) might be limited, I think that they are acceptable for a theoretical analysis.

---

> ### Author Rebuttal · Authors · 2025-03-31
>
> We thank the reviewer for their constructive comments. Below, we address each point in detail.
>
> > **(1) Tasks that were previously unlearnable without CoT (2) The method also appears to have an advantage...**
>
> We thank the reviewer for raising this question regarding whether the difference between CoT and non-CoT models is primarily an optimization issue or a generalization issue. To clarify, in our experiments, models trained without CoT are able to reduce the training loss to near-zero and achieve nearly 100% training accuracy. For clarity, we include an additional plot showing the training dynamics corresponding to Figure 1 of our main paper: [Link](https://files.catbox.moe/95rhrv.png). As shown, while the training accuracy approaches 100%, the test accuracy remains at random chance. This indicates that the key challenge is not optimization, but generalization: models trained without CoT fail to generalize to unseen compositions of subtasks, despite perfectly fitting the training data. In contrast, models trained with CoT exhibit strong generalization.
>
> The core contribution of our work is a shift in perspective from traditional sample complexity to task complexity. We show that for tasks with compositional structure—where CoT prompting serves as an example—training on a small set of tasks enables generalization to an exponentially large number of unseen tasks. Prior work [Wen et al (2024)] shows that transformers can, in principle, learn sparse parity functions without CoT, but doing so requires exponentially more data. In our setting, which focuses on in-context learning (ICL), we observe that even when the model fits the training tasks without CoT, it fails to generalize across tasks. In contrast, CoT enables generalization to an exponentially large space of unseen compositions, in line with our theoretical predictions. We will make this distinction clearer in the revision.
>
> > **Multirun experiment.**
>
> We appreciate the concern. It is worth noting that as we are focusing on task generalization, each time we change the number of tasks we need to rerun the experiments from scratch. In other words, each data point in our plot corresponds to a completely new random sample of tasks, which already provides an implicit robustness check. Nevertheless, in the revised version, we further strengthen our results by including additional experiments with repeated runs (at least 5 random seeds) for selected settings—specifically for $d=15$ and $d=20$—to directly assess the stability of our results. A plot summarizing the variance across seeds for these settings is available [here](https://files.catbox.moe/acfo1o.png).
>
> > **Essential References Not Discussed**
>
> We thank the reviewer for highlighting Task Diversity Shortens the ICL Plateau, which shows transformers can learn sparse parity when trained on a diverse task mix (e.g., parity with linear or quadratic regression). However, it's important to note that their setting differs from ours as they do not separate training tasks from testing tasks. Specifically, the model is trained continuously in an online fashion, where all tasks are repeatedly visited during training. In contrast, our work explicitly evaluates out-of-distribution task generalization by testing models on entirely unseen tasks. The key message of our work is that the ARC structure enables generalization to an exponentially large space of unseen compositions. Moreover, in their Figure 1, sparse parity (red line) isn’t learned on its own with transformers—it only becomes learnable when mixed with other task.
>
> > **How is the input prompt structured when training with CoT?**
>
> The input and overall setting are exactly the same as in the paper you mentioned, *“Understanding In-Context Learning…”*. The only difference is that, instead of providing only the final parity label, each label now includes the intermediate XOR computations (see the figure in the right column of page 5). For example, if $d = 15$, context length is 40, and $k = 3$, then the input size becomes $40 \times (15 + 3)$, compared to $40 \times (15 + 1)$ without CoT—just as in the original paper. As a result, zero-padding is not needed.
>
> > **Permutation-invariant**
>
> That's an interesting question. As the task has the same compositional structure, our theoretical framework still applies. However, as you noted, the number of possible tasks increases by a factor of $k!$, so the constants in the theory may differ. We ran experiments with $d = 15$ and $k = 3$, varying the number of training tasks. The results are summarized in the table in this [link](https://files.catbox.moe/63l1hh.png). (The number of training tasks is $n \cdot 15 \cdot \ln(15)$, where $n = 2, 3, 4$, and the total number of possible tasks is ${15 \choose 3} \cdot 3! = 2730$.) As expected, the generalization behavior roughly follows the pattern observed in the non-permutation case. However, we found that training was more difficult and required more steps to converge.

---

### Official Review · Reviewer_7pHd · 2025-03-13

**Overall Recommendation:** 4

**Summary:**

This paper presents the simple but useful and intuitive idea that *chain-of-thought generation reduces the theoretical complexity of compositional problems, and therefore leads to theoretically and empirically less required samples for strong generalisation* to unseen functions. Specifically, prior work has shown that transformers fail with this type of generalisation, but the authors of this work show they can in fact do it. The question the authors investigate is when can transformers generalise to a large number of unseen tasks from a smaller number of tasks? They show for an autoregressive compositionally structured task formulation how many tasks a model needs to theoretically see to generalise to all heldout tasks, and empirically demonstrate that this holds for three examples of such tasks. Even when task space gets larger (e.g. through more steps in the sequence), the amount of tasks needed to generalise follows the theoretically found scaling law. The authors further show that this is only the case when the model is trained on i.i.d. tasks and examples of these tasks, and has worse generalisation performance when specific examples/tasks are structurally held-out.

## Update after rebuttal

My points have been addressed after the rebuttal and I am still in favour of accepting this paper.

**Claims And Evidence:**

All claims are supported by clear and convincing evidence. The authors derive a theoretical result and show it empirically holds for three separate tasks for which the critical assumption in the theory holds.

This is not central to the thesis of the work, but the claim that is not supported is the following. In Section 5.2.2. the authors show results on a toy word translation task, and claim that *"This example illustrates that autoregressive compositional structures naturally arise in real-world languages, even without explicit CoT."* But this is a bit of a stretch, given that sequential translation of words to different languages is not a real-world language task.

One suggestion I would make to make the evidence stronger is to show that the scaling law does not hold for a task that violates the assumption about compositional identifiability.

**Essential References Not Discussed:**

Some papers the authors could consider citing are the following (just a suggestion, I leave it up to the authors if they are worth discussing):
- https://arxiv.org/abs/2307.03381 (related in that they show CoT is crucial for strong generalisation)
- https://arxiv.org/abs/2406.02550 (related in that they focus on task generalisation)

**Experimental Designs Or Analyses:**

The experimental design is sound and valid.

**Methods And Evaluation Criteria:**

All methods and evaluation criteria make sense.

**Other Comments Or Suggestions:**

List of typos:
- 117r typo finte -> finite
- 315r typo ambien -> ambient
- 346-348l needs a rewrite (mostly in terms of verbs and determiners)
- 365-366l as well
- 111r typo hide -> hides
- 193l "can only access to" -> "can only access"
- 202l "it accesses to" -> "it has access to"
- 211r "two stage" -> "two stages"
- Typo title figure 3 arithmatic -> arithmetic

**Other Strengths And Weaknesses:**

**Strengths**
- Very clear writing
- Neat connection of theory to multiple empirical results
- The idea presented here contributed to my understanding of generalisation in transformers


**Weaknesses**
- No real-world tasks
- As mentioned above, would be interesting to show a failure case (different number of tasks required than predicted by theory for a task that violates compositional identifiability)

**Questions For Authors:**

- Do you have any idea why the compounding of errors does not show up in the other tasks than the word translation one?

**Relation To Broader Scientific Literature:**

The authors successfully demonstrate how transformers can show strong generalisation in cases that prior work said they struggle with, and come up with a new theoretical result that explains why and how, which can potentially connect to generalisation properties of LLMs.

**Theoretical Claims:**

I did not check the correctness of any proofs and theoretical claims, but the empirical results validate them and they make intuitively perfect sense. E.g. if a problem is compositional, in that each next step to tackle is independent of the previous steps if you keep track of a state, chain-of-thought can break down the problem and leads to a lower complexity task. Generalisation then is independent of the overall task complexity in parameters like sequence length cause one can compositionally apply the algorithm to each step.

---

> ### Author Rebuttal · Authors · 2025-03-31
>
> We sincerely appreciate the reviewer's support and valuable suggestions. Below, we address each point in detail.
>
> > The sequential translation task is not a real-world language task.
>
> Thank you for pointing this out. We agree the translation example is synthetic and will clarify that our goal was merely to illustrate how decompositional structure can emerge without explicit CoT, as long as compositional structure holds, rather than to suggest that this task reflects real-world language use. We will clarify this in the final version.
>
> >  A failure case where the compositional identifiability assumption is violated so that the scaling law fails.
>
> Thank you for the suggestion. At a high level, the compositional identifiability assumption is required to ensure that any subtask is identifiable regardless of the other subtasks it is composed with. To illustrate what happens when this assumption is violated, we present an extreme failure case where a subtask is identifiable only in the presence of specific preceding subtasks. In this setting, task complexity grows exponentially with $T$:
>
> Let the vocabulary be $\mathcal{Y} = \{0, 1\}$. Consider the following autoregressive compositional task class where the output is deterministic and independent of the input $x$.
>
> For timesteps $1 \leq t \leq T-1$, there are two possible tasks:
>
> - $\theta_t^0$: Always output $y_t = 0$, regardless of $(x, y_{<t})$.
> - $\theta_t^1$: Always output $y_t = 1$, regardless of $(x, y_{<t})$.
>
> At timestep $t = T$, there are also two tasks:
>
> - $\theta_T^0$: Output $y_T = y_1 \land y_2 \land \cdots \land y_{T-1},$ where $\land$ denotes the AND operation.
>
> - $\theta_T^1$: Always output $y_T = 1$, regardless of $(x, y_{<T})$.
>
> In this case, identifying the subtask $\theta_T^0$ as distinct from simply outputting $y_T = 0$ requires that the specific task $(\theta_1^1, \theta_2^1, \dots, \theta_{T-1}^1, \theta_T^0)$ appears in training. The probability of sampling this task decreases exponentially with $T$, making the task complexity exponentially large for successful learning. We will briefly discuss this in the final version.
>
>
> > Why the compounding of errors does not show up in the other tasks than the word translation one
>
> This is a great and challenging question. One plausible explanation is that the implicit biases of Transformers vary significantly across tasks. In the word translation task, the model may adopt undesirable shortcuts—such as only partially solving the intended task (e.g., learning to translate from Chinese to French without generalizing from English to Chinese)—which increases the likelihood of errors at each generation step. In autoregressive generation, an error rate of $p$ per step compounds over time, meaning that achieving correct sequence generation with constant probability requires $p = o(1/T)$. In contrast, algorithmic tasks like parity and arithmetic appear to induce stronger biases toward learning the correct underlying computation, potentially mitigating this issue. That said, we stress that this compounding error argument is merely a hypothesis and does not constitute a rigorous explanation. We do not fully understand why these tasks exhibit different scaling behaviors, and we leave a complete explanation for future work.
>
> > No real-world tasks.
>
> Our goal is to establish a foundation where the compositional structure of tasks is clearly defined and generalization to unseen tasks is fully controllable. To this end, we deliberately focus on synthetic tasks to isolate the core principles underlying ARC-style generalization, explicitly specifying seen tasks and rigorously evaluating generalization on truly unseen tasks, free from leakage or contamination from pretraining.
>
> We also note that several influential works studying transformer behaviors similarly focus on synthetic or simplified settings, such as parity functions [1] and modular arithmetic [2,3], which have proven effective for uncovering key insights.
>
> While extending this framework to broader NLP tasks is important, it incurs substantial computational cost, which is beyond the scope of this paper. We view our current results as a necessary first step and are actively exploring such extensions.
>
> [1] Bhattamishra et al., *Understanding in-context learning in transformers and LLMs by learning to learn discrete functions*, ICLR 2024.
> [2] Power et al., *Grokking: Generalization Beyond Overfitting on Small Algorithmic Datasets*, 2022.
> [3] Nanda et al., *Progress measures for grokking via mechanistic interpretability*, ICLR 2023.
>
> > Typos and Related Papers in the paper.
>
> We thank the reviewer for the careful feedback and suggestions. We will correct the typos and include the suggested reference in the final version of the paper.

---

### Official Review · Reviewer_vSqF · 2025-03-13

**Overall Recommendation:** 3

**Summary:**

This paper investigates when models trained on a small set of tasks can generalize to a much larger task family. The authors approach this through the lens of "autoregressive compositional structure" (ARC), where tasks are composed of T sequential operations, each chosen from D possible subtasks, creating a total task space of D^T. The key theoretical contribution is proving that generalization to all D^T tasks can be achieved by training on only O(D log DT) tasks. Empirically, the authors demonstrate that Transformers can achieve this exponential generalization on sparse parity functions and arithmetic operations.

**Claims And Evidence:**

1. The theoretical claims are supported by the proof.
2. The empirical claim on the dependency of the function space (D) and the number of compositions (T) are not sufficiently convincing. First, for each task, only a handful number of D and T choices are tested, which is still relatively small to verify the bound.

**Essential References Not Discussed:**

Most recent works in this area that I am familar with are cited.

**Experimental Designs Or Analyses:**

1. About the experimental setup: There is no detail on the arithmetic task. Is it still following the ICL format and what does it look like?
2. About the empirical results: Besides the number of tested D and T is small, the visualizations to showcase the dependency of the number of training tasks are also relatively confusing. The current plots have test acc (the y axis) vs num of training tasks/Dlog(DT) (the x axis) which is hard to see whether it scales in a linear/log way. More natual visualizations would be number of training tasks used to achieve ~100% test acc vs Dlog(DT)), with the actual linear/log line as a reference.

**Methods And Evaluation Criteria:**

Yes the proposed methods and/or evaluation criteria (e.g., benchmark datasets) make sense for the problem or application at hand.

**Other Comments Or Suggestions:**

1. I recommend making the experimental section more comprehensive with clearer visualizations for the linear/log dependency.
2. The paper would benefit from more details on experimental setups for each task.

**Other Strengths And Weaknesses:**

Strengths:
1.  The paper introduces a structured mathematical framework that quantifies compositional complexity through parameters D and T and provide theoretical result for the sample complexity.
2. The paper designs ICL experiments to verify the theortical results.

Weaknesses:
1. The dimensions tested are relatively small compared to the parameter spaces of modern LLMs. The generalization properties might behave differently at much larger scales.
2. The experiments focus on relatively synthetic tasks with clear compositional structure. It remains unclear how these findings might transfer to more complex, natural language tasks.

**Questions For Authors:**

See above.

**Relation To Broader Scientific Literature:**

The key theoretical contribution is proving that generalization to all D^T tasks can be achieved by training on only O(D log DT) tasks. I fell like this result is relatively new but it would be great if the author could clarify that how the proposed frame work and the theoretical result is different from the compositional generalization literature. For the empirical work, though the authors claim that it verifies the theory, I still feel like they're not very convincing for the reasons mentioned in Claims And Evidence & Experimental Designs Or Analyses.

**Theoretical Claims:**

1. I briefly read the proof but didn't check the details. One confusing thing is that it seems to be more like a general approach rather than being specific to the autoregressive prediction, which is probably also the reason that the authors observe a different dependency on T in the language translation task.
2. In the paper, it assumes the number of demonstration (nx) goes to infinity which seems not very realistic.

---

> ### Author Rebuttal · Authors · 2025-03-31
>
> We thank the reviewer for their comments and valuable suggestions. Below, we address each point in detail.
>
> >Dependency on $d$ and $k$ in Experiments
>
> We agree that a broader range of $d$ and $k$ provides additional support for our findings. However, both $d$ and $k$ appear in the base and exponent when determining the total number of tasks, leading to exponential growth. For instance, in the parity task, even with $d=25$ and $k=10$, the number of distinct parity functions is $\binom{25}{10} \approx 3$ million. We have strengthened our empirical support by adding this new experiment on parity tasks with $d=25$ and $k=10$:
>
> | $d$ | $k$ | \# Training Tasks | \# Total Tasks | Accuracy (%) |
> |---|---|-|-|-|
> | 10 | 5 | 69 | 252 | 98.51 |
> | 15 | 7 | 121 | 6,400 | 99.12 |
> | 25 | 10 | 241 | 3.2M | 98.60 |
>
> These results clearly indicate that the number of training tasks grows approximately linearly while the total number of tasks grow exponentially.
>
> It is still a valid question to ask why can’t we scale to significantly larger $d$. The main challenge comes from context length. Unlike prior works focusing on learning a fixed parity function where $d$ simply reflected input dimensionality, we are in an ICL setup where the model needs to learn a general algorithm that infers *any* parity function drawn from a large function family. For the problem to be identifiable, the number of in-context examples $N$ must scale linearly with $d$. For example, when $d=100$ and $k=50$, $N=100$ examples result in an input length of roughly $N(d+k) \approx 15,000$ tokens -- which surpasses our available computational resources. This quadratic scaling of context length is the main bottleneck limiting us from larger $d$ and $k$.
>
> >Theoretical Claims and Autoregressive Specificity
>
> While the proof may appear general, our framework is specific to the autoregressive (ARC) setting. The distribution of $y_t$ explicitly depends on $y_{<t}$, and our constructed learner solves each subtask autoregressively, only using $(x,y_{\leq t})$ from the input sequence. Setups without ARC structure can be seen as a special case (when $T=1$), but compositionality, which is central to our analysis, only emerges through ARC composition when $T>1$.
>
> >Assumption of Infinite Demonstrations
>
> The assumption $n_x\to\infty$ in the main text is for simplifying exposition. We also provide a non-asymptotic guarantee in Appendix B (Remark 3.4, Theorem B.2). Specifically, if each subtask distribution is separated from incorrect hypotheses by a total-variation margin $\epsilon>0$, then $n_x=\widetilde{O}(1/\epsilon^2)$ demonstrations suffice. Thus, our analysis covers the finite-sample regime.
>
> >Visualization of Experimental Results
>
> The visualization suggested by the reviewer—plotting the number of training tasks needed to reach a given accuracy—is a valid alternative. However, we believe there is no single best way to present the results. Our current presentation is chosen to align closely with the theoretical prediction on task complexity. It also provides both lower and upper bounds for a given accuracy. For instance, Figure 1 shows that in the parity task, all setups achieve ≥98% accuracy within $[2D\log D,4D\log D]$ training tasks.
>
> >Relation to Prior Literature on Compositional Generalization
>
> Our framework differs from prior work in two key ways. First, most existing work is largely empirical, focusing on assessing compositional abilities of pretrained LMs, while we explicitly study how compositionality emerges during training when structured task families are presented. Second, we shift the focus from standard sample complexity to *task complexity*, aiming for generalization to unseen tasks where training on a small number of tasks generalizes to exponentially many unseen tasks.
>
> >Experimental Details: Arithmetic Task
>
> The arithmetic task fully follows the in-context learning (ICL) setting, consistent with the parity task. We will add more experimental details in the final version.
>
> >Use of Synthetic Tasks
>
> We chose synthetic tasks to ensure that the compositional structure is fully controlled and interpretable. The goal is to establish a foundation where the compositional structure is clearly defined, and generalization to unseen tasks is fully controllable. We also note that several influential works studying transformer behaviors similarly focus on synthetic or simplified settings, such as parity functions [1] and modular arithmetic [2,3], which have proven effective for uncovering key insights. In our case, the key insight is *exponential task generalization*.
>
> Extending this framework to real-world NLP tasks is important and an active direction we are exploring. However, doing so requires substantial more work, which we believe deserves a dedicated follow-up study.
>
> [1] Bhattamishra et al. (https://arxiv.org/abs/2310.03016 )
>
> [2] Power et al. (https://arxiv.org/abs/2201.02177)
>
> [3] Nanda et al. (https://arxiv.org/abs/2301.05217)

---

> > ### Comment · Reviewer_vSqF · 2025-04-03
> >
> > Thank you for the response. I think the question "About the experimental setup: There is no detail on the arithmetic task. Is it still following the ICL format and what does it look like?" is not answered. Could you verify?

---

> > > ### Author Response · Authors · 2025-04-03
> > >
> > > We briefly mentioned in the second-to-last response, the arithmetic task indeed follows the same in-context learning (ICL) setup as the parity task, and we are happy to provide more details here.
> > >
> > > To elaborate, consider the case where the input dimension is d = 5. Each task consists of an operation sequence of length d - 1 = 4, represented by a tuple of arithmetic operations (either `+` or `*`), such as (`+`, `+`, `*`, `+`). A sample input could be the binary vector `11110`.
> > >
> > > The corresponding chain-of-thought (CoT) reasoning involves applying the operations step-by-step, like this:
> > >
> > > - 1 + 1 = 2  — Step 1 of CoT
> > > - (1 + 1) + 1 = 3  — Step 2 of CoT
> > > - ((1 + 1) + 1) * 1 = 3  — Step 3 of CoT
> > > - (((1 + 1) + 1) * 1) + 0 = 3  — Step 4 (final answer)
> > >
> > >
> > > So the CoT steps are `2, 3, 3`, and the final step gives the answer `3`. The final String looks like this $11110\rightarrow\underbrace{233}_{\text{CoT steps}}3.$
> > > The rest follows the same ICL format as the figure on page 5, identical to the parity setup, i.e. we concatenate \(N\) such examples as in-context demonstrations, followed by a query input, and ask the model to produce the query's CoT steps and final answer.
> > >
> > > We appreciate your thoughtful feedback and hope that our response has addressed your concerns and helped shift your assessment in a positive direction. We’re happy to clarify further during the rebuttal window if needed.

---

### Official Review · Reviewer_B7oU · 2025-03-18

**Overall Recommendation:** 3

**Summary:**

This paper investigates task generalization in large language models (LLMs) through the lens of AutoRegressive Compositional (ARC) structure. The central question explored is: When can learning from a small set of tasks enable generalization to a much larger task family? The authors propose that LLMs, particularly transformers, can achieve exponential task generalization when tasks are structured compositionally.

**Claims And Evidence:**

The paper makes several claims that are appropriately supported by empirical results.

Claim: Generalization to exponentially large task sets is possible with limited training tasks when tasks follow ARC structure.
- Empirical results in sparse parity and arithmetic tasks show that transformers

Claim: Task selection significantly impacts generalization.
- When tasks are sampled adversarially (e.g., omitting tasks with a specific coordinate value), transformers fail to generalize even when trained on an otherwise large task set.

**Essential References Not Discussed:**

I am not aware of essential references not discussed

**Experimental Designs Or Analyses:**

The experimental setup is well-designed, focusing on synthetic benchmarks that allow controlled evaluation of task generalization.
(See Methods and Evaluation Criteria part).

**Methods And Evaluation Criteria:**

Yes, their proposed framework and the evaluation criteria make sense and support their theroem.
More specifically,
- the ARC framework is clearly defined and analyzed mathematically
- the experimental setup isolates core factors influencing generalization (number of secret indices, ambient dimension, task selection, CoT usage)
- uses linear probing for subtask identification.

**Other Comments Or Suggestions:**

- (minor) Typo in line 933 Wadam -> adamW?
- model scale experiments (rather than with 3-layer, 1-head model) could enhance the empirical results.

**Other Strengths And Weaknesses:**

### Strengths
- Theoretical contributions are rigorous, providing a well-defined framework for studying task generalization.
- Scaling laws are empirically validated, demonstrating consistency between theory and practice.
- Task selection ablations are insightful, highlighting the importance of diverse training examples.

### Weaknesses
- Scope is limited to synthetic tasks, making it unclear whether the results translate to broader NLP applications.
- Lacks model scale experiments
- The range of k (number of secret indices), and d (ambient dimensions) is narrow.

**Questions For Authors:**

- How does task generalization change with model scale?
- Would pretraining on structured reasoning datasets improve ARC generalization?

**Relation To Broader Scientific Literature:**

The paper connects well to prior work in:
- Compositional generalization in neural networks
- In-context learning and task transferability
- CoT reasoning and its impact on LLM expressiveness

**Theoretical Claims:**

Yes. I checked Theorem 3.3 and the proof in the supplementary material seems correct.

---

> ### Author Rebuttal · Authors · 2025-03-31
>
> We thank the reviewer for their positive and constructive review. Below, we address each point in detail.
>
> > Scope is limited to synthetic tasks
>
> Our goal is to establish a foundation where the compositional structure of tasks is clearly defined and generalization to unseen tasks is fully controllable. To this end, we deliberately focus on synthetic tasks to isolate the core principles underlying ARC-style generalization, explicitly specifying seen tasks and rigorously evaluating generalization on truly unseen tasks, free from leakage or contamination from pretraining.
>
> We also note that several influential works studying transformer behaviors similarly focus on synthetic or simplified settings, such as parity functions [1] and modular arithmetic [2,3], which have proven effective for uncovering key insights. In our case, the key insight is *exponential task generalization*: the model generalizes to a combinatorially large set of unseen tasks after training on only a small subset.
>
> While extending this framework to broader NLP tasks is important, it incurs substantial computational cost, which is beyond the scope of this paper. We view our current results as a necessary first step and are actively exploring such extensions.
>
> [1] Bhattamishra et al., *Understanding in-context learning in transformers and LLMs by learning to learn discrete functions*, ICLR 2024.
> [2] Power et al., *Grokking: Generalization Beyond Overfitting on Small Algorithmic Datasets*, 2022.
> [3] Nanda et al., *Progress measures for grokking via mechanistic interpretability*, ICLR 2023.
>
>
> >The range of $d$ and $k$ is narrow
>
> We agree that a broader range of $d$ and $k$ provides additional support for our findings. However, both $d$ and $k$ appear in the base and exponent when determining the total number of tasks, leading to exponential growth. For instance, in the parity task, even with $d=25$ and $k=10$, the number of distinct parity functions is $\binom{25}{10} \approx 3$ million. We have strengthened our empirical support by adding this new experiment on parity tasks with $d=25$ and $k=10$,
>
> | $d$ | $k$ | \# Training Tasks | \# Total Tasks | Accuracy (%) |
> |-----|-----|------------------|----------------|--------------|
> | 10 | 5 | 69 | 252 | 98.51% |
> | 15 | 7 | 121 | 6,400 | 99.12% |
> | 25 | 10 | 241 | 3.2M | 98.60% |
>
> It is still a valid question to ask why can’t we scale to significantly larger $d$. The main challenge comes from context length. Unlike prior works focusing on learning a fixed parity function where $d$ simply reflected input dimensionality, we are in an ICL setup where the model needs to learn a general algorithm that infers *any* parity function drawn from a large function family. For the problem to be identifiable, the number of in-context examples $N$ must scale linearly with $d$. For example, when $d=100$ and $k=50$, $N=100$ examples result in an input length of roughly $N(d+k) \approx 15,000$ tokens -- which surpasses our available computational resources. This quadratic scaling of context length is the main bottleneck limiting us from larger $d$ and $k$.
>
>
> >Lack of model scale experiments
>
> In early experiments, we varied the number of layers and heads. We observed that while increasing layers from 2 to 3 improved generalization, going beyond 3 layers yielded no additional benefit. Similarly, increasing attention heads did not improve performance. This motivated our choice of a 3-layer, 1-head model for faster optimization without sacrificing generalization. A supporting plot from our early experiments is available in this [link](https://files.catbox.moe/x1r11m.png).
>
> We also conducted a new set of experiments comparing models with width 192 vs. 384, 3 layers vs. 6 layers, and 1 head vs. 4 heads, across different numbers of training tasks (120 and 160), with fixed input dimension \(d = 15\) and \(k = 3\). The results were consistent, showing comparable performance across configurations, in this table [Link](https://files.catbox.moe/dlsa4k.png).
>
> Importantly, our model already exhibits the near-linear task complexity predicted by theory. Larger models would not meaningfully improve this scaling.
>
>
> >Would pretraining on structured reasoning datasets improve ARC generalization?
>
> This is an excellent question. Recent work on R1 distillation has shown that strong generalization can be achieved from a relatively small number of SFT samples [4,5]. Inspired by this, we hypothesize that pretraining on structured reasoning datasets could further enhance ARC generalization.
>
> However, this direction requires significant additional work and resources, as pretraining large models on diverse reasoning datasets is far more demanding than our current ICL setting. Nonetheless, we believe this is a promising avenue for future research.
>
> [4] Ye et al., *LIMO: Less is More for Reasoning*, 2025.
> [5] Muennighoff et al., *s1: Simple test-time scaling*, 2025.
>
>
> >Typo in line 933
>
> Thank you for pointing this out. We will fix it.

---

### Decision · Program_Chairs · 2025-05-01

**Decision:**

Accept (poster)

**Comment:**

This paper investigates transformer models' ability to generalize from a small set of tasks to an exponentially larger task family. The authors prove that generalization to D^T tasks can be achieved by training on only O(D log DT) tasks under the task identifiability assumption. (Note that O(D log DT) tasks imply O(DT log DT) 'subtasks', so one can learn the D subtasks for each time step.) The theoretical results are validated empirically across both synthetic tasks that satisfy the assumptions and those that don't.

The paper merits acceptance due to its rigorous theoretical foundation supported by consistent empirical validation. Reviewers appreciated the clear mathematical framework for the compositional structure, the demonstrated alignment between theoretical predictions and experimental results, the excellent writing quality, and the insightful ablations that highlighted the importance of diverse training examples.